# BENCHMARKS FOR DEEP OFF-POLICY EVALUATION

**Justin Fu**[*1]  **Mohammad Norouzi**[*2]  **Ofir Nachum**[*2]  **George Tucker**[*2]
**Ziyu Wang**[2]  **Alexander Novikov**[3]  **Mengjiao Yang**[2]  **Michael R. Zhang**[2]
**Yutian Chen**[3]  **Aviral Kumar**[1]  **Cosmin Paduraru**[3]  **Sergey Levine**[1]  **Tom Le Paine**[*3]

[1]UC Berkeley  [2]Google Brain  [3]DeepMind
justinfu@berkeley.edu,{mnorouzi,ofirnachum,gjt,tpaine}@google.com

## ABSTRACT

Off-policy evaluation (OPE) holds the promise of being able to leverage large, offline datasets for both evaluating and selecting complex policies for decision making. The ability to learn offline is particularly important in many real-world domains, such as in healthcare, recommender systems, or robotics, where online data collection is an expensive and potentially dangerous process. Being able to accurately evaluate and select high-performing policies without requiring online interaction could yield significant benefits in safety, time, and cost for these applications. While many OPE methods have been proposed in recent years, comparing results between papers is difficult because currently there is a lack of a comprehensive and unified benchmark, and measuring algorithmic progress has been challenging due to the lack of difficult evaluation tasks. In order to address this gap, we present a collection of policies that in conjunction with existing offline datasets can be used for benchmarking off-policy evaluation. Our tasks include a range of challenging high-dimensional continuous control problems, with wide selections of datasets and policies for performing policy selection. The goal of our benchmark is to provide a standardized measure of progress that is motivated from a set of principles designed to challenge and test the limits of existing OPE methods. We perform an evaluation of state-of-the-art algorithms and provide open-source access to our data and code to foster future research in this area[†].

## 1 INTRODUCTION

Reinforcement learning algorithms can acquire effective policies for a wide range of problems through active online interaction, such as in robotics (Kober et al., 2013), board games and video games (Tesauro, 1995; Mnih et al., 2013; Vinyals et al., 2019), and recommender systems (Aggarwal et al., 2016). However, this sort of active online interaction is often impractical for real-world problems, where active data collection can be costly (Li et al., 2010), dangerous (Hauskrecht & Fraser, 2000; Kendall et al., 2019), or time consuming (Gu et al., 2017). Batch (or offline) reinforcement learning, has been studied extensively in domains such as healthcare (Thapa et al., 2005; Raghu et al., 2018), recommender systems (Dudík et al., 2014; Theocharous et al., 2015; Swaminathan et al., 2017), education (Mandel et al., 2014), and robotics (Kalashnikov et al., 2018). A major challenge with such methods is the off-policy evaluation (OPE) problem, where one must evaluate the expected performance of policies solely from offline data. This is critical for several reasons, including providing high-confidence guarantees prior to deployment (Thomas et al., 2015), and performing policy improvement and model selection (Bottou et al., 2013; Doroudi et al., 2017).

The goal of this paper is to provide a standardized benchmark for evaluating OPE methods. Although considerable theoretical (Thomas & Brunskill, 2016; Swaminathan & Joachims, 2015; Jiang & Li, 2015; Wang et al., 2017; Yang et al., 2020) and practical progress (Gilotte et al., 2018; Nie et al., 2019; Kalashnikov et al., 2018) on OPE algorithms has been made in a range of different domains, there are few broadly accepted evaluation tasks that combine complex, high-dimensional problems

---

[*]Equally major contributors.
[†]Policies and evaluation code are available at https://github.com/google-research/deep_ope. See Section 5 for links to modelling code.

commonly explored by modern deep reinforcement learning algorithms (Bellemare et al., 2013; Brockman et al., 2016) with standardized evaluation protocols and metrics. Our goal is to provide a set of tasks with a range of difficulty, excercise a variety of design properties, and provide policies with different behavioral patterns in order to establish a standardized framework for comparing OPE algorithms. We put particular emphasis on large datasets, long-horizon tasks, and task complexity to facilitate the development of scalable algorithms that can solve high-dimensional problems.

Our primary contribution is the **D**eep **O**ff-**P**olicy **E**valuation (DOPE) benchmark. DOPE is designed to measure the performance of OPE methods by **1)** evaluating on challenging control tasks with properties known to be difficult for OPE methods, but which occur in real-world scenarios, **2)** evaluating across a range of policies with different values, to directly measure performance on policy evaluation, ranking and selection, and **3)** evaluating in ideal and adversarial settings in terms of dataset coverage and support. These factors are independent of task difficulty, but are known to have a large impact on OPE performance. To achieve 1, we selected tasks on a set of design principles outlined in Section 3.1. To achieve 2, for each task we include 10 to 96 policies for evaluation and devise an evaluation protocol that measures policy evaluation, ranking, and selection as outlined in Section 3.2. To achieve 3, we provide two domains with differing dataset coverage and support properties described in Section 4. Finally, to enable an easy-to-use research platform, we provide the datasets, target policies, evaluation API, as well as the recorded results of state-of-the-art algorithms (presented in Section 5) as open-source.

## 2 BACKGROUND

We briefly review the off-policy evaluation (OPE) problem setting. We consider Markov decision processes (MDPs), defined by a tuple $(\mathcal{S}, \mathcal{A}, \mathcal{T}, R, \rho_0, \gamma)$, with state space $\mathcal{S}$, action space $\mathcal{A}$, transition distribution $\mathcal{T}(s'|s, a)$, initial state distribution $\rho_0(s)$, reward function $R(s, a)$ and discount factor $\gamma \in (0, 1]$. In reinforcement learning, we are typically concerned with optimizing or estimating the performance of a policy $\pi(a|s)$.

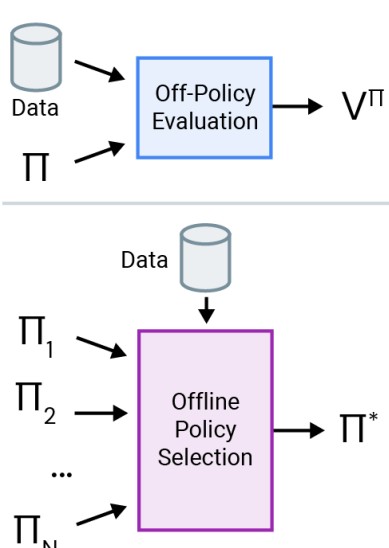

The performance of a policy is commonly measured by the *policy value* $V^\pi$, defined as the expected sum of discounted rewards:

$$V^\pi := \mathbb{E}_{s_0 \sim \rho_0, s_{1:\infty}, a_{0:\infty} \sim \pi} \left[ \sum_{t=0}^{\infty} \gamma^t R(s_t, a_t) \right]. \quad (1)$$

If we have access to state and action samples collected from a policy $\pi$, then we can use the sample mean of observed returns to estimate the value function above. However, in off-policy evaluation we are typically interested in estimating the value of a policy when the data is collected from a separate *behavior policy* $\pi_B(a|s)$. This setting can arise, for example, when data is being generated online from another process, or in the purely offline case when we have a historical dataset.

Figure 1: In Off-Policy Evaluation (top) the goal is to estimate the value of a single policy given only data. Offline Policy Selection (bottom) is a closely related problem: given a set of N policies, attempt to pick the best given only data.

In this work we consider the latter, purely offline setting. The typical setup for this problem formulation is that we are provided with a discount $\gamma$, a dataset of trajectories collected from a behavior policy $\mathcal{D} = \{(s_0, a_0, r_0, s_1, \ldots)\}$, and optionally the action probabilities for the behavior policy $\pi_B(a_t|s_t)$. In many practical applications, logging action propensities is not possible, for example, when the behavior policy is a mix of ML and hard-coded business logic. For this reason, we focus on the setting without propensities to encourage future work on behavior-agnostic OPE methods. For the methods that require propensities, we estimate the propensities with behavior cloning.

The objective can take multiple flavors, as shown in Fig. 1. A common task in OPE is to estimate the performance, or value, of a policy $\pi$ (which may not be the same as $\pi_B$) so that the estimated

value is as close as possible to $V^\pi$ under a metric such as MSE or absolute error. A second task is to perform policy selection, where the goal is to select the best policy or set of policies out of a group of candidates. This setup corresponds to how OPE is commonly used in practice, which is to find the best performing strategy out of a pool when online evaluation is too expensive to be feasible.

# 3 DOPE: DEEP OFF-POLICY EVALUATION

The goal of the Deep Off-Policy Evaluation (DOPE) benchmark is to provide tasks that are challenging and effective measures of progress for OPE methods, yet is easy to use in order to better facilitate research. Therefore, we design our benchmark around a set of properties which are known to be difficult for existing OPE methods in order to gauge their shortcomings, and keep all tasks amenable to simulation in order for the benchmark to be accessible and easy to evaluate.

## 3.1 TASK PROPERTIES

We describe our motivating properties for selecting tasks for the benchmark as follows:

**High Dimensional Spaces (H)** High-dimensionality is a key-feature in many real-world domains where it is difficult to perform feature engineering, such as in robotics, autonomous driving, and more. In these problems, it becomes challenging to accurately estimate quantities such as the value function without the use of high-capacity models such a neural networks and large datasets with wide state coverage. Our benchmark contains complex continuous-space tasks which exercise these challenges.

**Long Time-Horizon (L)** Long time horizon tasks are known to present difficult challenges for OPE algorithms. Some algorithms have difficulty doing credit assignment for these tasks. This can be made worse as the state dimension or action dimension increases.

**Sparse Rewards (R)** Sparse reward tasks increase the difficulty of credit assignment and add exploration challenges, which may interact with data coverage in the offline setting. We include a range robotics and navigation tasks which are difficult to solve due to reward sparsity.

**Temporally extended control (T)** The ability to make decisions hierarchically is major challenge in many reinforcement learning applications. We include two navigation tasks which require high-level planning in addition to low-level control in order to simulate the difficulty in such problems.

## 3.2 EVALUATION PROTOCOL

The goal of DOPE to provide metrics for policy ranking, evaluation and selection. Many existing OPE methods have only been evaluated on point estimates of value such as MSE, but policy selection is an important, practical use-case of OPE. In order to explicitly measure the quality of using OPE for policy selection, we provide a set of policies with varying value, and devise two metrics that measure how well OPE methods can rank policies.

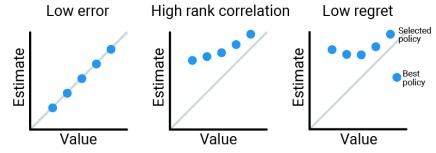

Figure 2: Error is a natural measure for off-policy evaluation. However for policy selection, it is sufficient to (i) rank the policies as measured by rank correlation, or (ii) select a policy with the lowest regret.

For each task we include a dataset of logged experiences $\mathcal{D}$, and a set of policies $\{\pi_1, \pi_2, ..., \pi_N\}$ with varying values. For each policy, OPE algorithms must use $\mathcal{D}$ to produce an estimate of the policy's value. For evaluation of these estimates, we provide "ground truth values" $\{V^{\pi_1}, V^{\pi_2}, ..., V^{\pi_N}\}$ that are computed by running the policy for $M \geq 1000$ episodes, where the exact value of $M$ is given by the number of episodes needed to lower the error bar on the ground truth values to $0.666$. The estimated values are then compared to these ground truth values using three different metrics encompassing both policy evaluation and selection (illustrated in Figure 2; see Appendix A.1 for mathematical definitions).

**Absolute Error** This metric measures estimate accuracy instead of its usefulness for ranking. Error is the most commonly used metric to assess performance of OPE algorithms. We opted to use absolute error instead of MSE to be robust to outliers.

**Regret@k** This metric measures how much worse the best policies identified by the estimates are than the best policy in the entire set. It is computed by identifying the top-k policies according to the estimated returns. Regret@k is the difference between the actual expected return of the best policy in the entire set, and the actual value of the best policy in the top-k set.

**Rank correlation** This metric directly measures how well estimated values rank policies, by computing the correlation between ordinal rankings according by the OPE estimates and ordinal rankings according to the ground truth values.

## 4 DOMAINS

DOPE contains two domains designed to provide a more comprehensive picture of how well OPE methods perform in different settings. These two domains are constructed using two benchmarks previously proposed for offline reinforcement learning: RL Unplugged (Gulcehre et al., 2020) and D4RL (Fu et al., 2020), and reflect the challenges found within them.

The **DOPE RL Unplugged** domain is constrained in two important ways: 1) the data is always generated using online RL training, ensuring there is adequate coverage of the state-action space, and 2) the policies are generated by applying offline RL algorithms to the same dataset we use for evaluation, ensuring that the behavior policy and evaluation policies induce similar state-action distributions. Using it, we hope to understand how OPE methods work as task complexity increases from simple Cartpole tasks to controlling a Humanoid body while controlling for ideal data.

On the other hand, the **DOPE D4RL** domain has: 1) data from various sources (including random exploration, human teleoperation, and RL-trained policies with limited exploration), which results in varying levels of coverage of the state-action space, and 2) policies that are generated using online RL algorithms, making it less likely that the behavior and evaluation policies share similar induced state-action distributions. Both of these result in distribution shift which is known to be challenging for OPE methods, even in simple tasks. So, using it we hope to measure how well OPE methods work in more practical data settings.

### 4.1 DOPE RL UNPLUGGED

DeepMind Control Suite (Tassa et al., 2018) is a set of control tasks implemented in MuJoCo (Todorov et al., 2012). We consider the subset included in RL Unplugged. This subset includes tasks that cover a range of difficulties. From Cartpole swingup, a simple task with a single degree 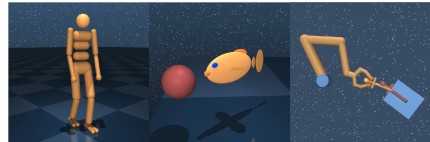 of freedom, to Humanoid run which involves control of a complex bodies with 21 degrees of freedom. All tasks use the default feature representation of the system state, including proprioceptive information such as joint positions and velocity, and additional sensor information and target position where appropriate. The observation dimension ranges from 5 to 67.

**Datasets and policies** We train four offline RL algorithms (D4PG (Barth-Maron et al., 2018), ABM (Siegel et al., 2020), CRR (Wang et al., 2020) and behavior cloning), varying their hyperparameters. For each algorithm-task-hyperparameter combination, we train an agent with 3 random seeds on the DM Control Suite dataset from RL Unplugged and record policy snapshots at exponentially increasing intervals (after 25k learner steps, 50k, 100K, 200K, etc). Following Gulcehre et al. (2020), we consider a deterministic policy for D4PG and stochastic policies for BC, ABM and CRR. The datasets are taken from the RL Unplugged benchmark, where they were created by training multiple (online) RL agents and collecting both successful and unsuccessful episodes throughout training. All offline RL algorithms are implemented using the Acme framework (Hoffman et al., 2020).

### 4.2 DOPE D4RL

**Gym-MuJoCo tasks.** Gym-MuJoCo consists of several continuous control tasks implemented within the MuJoCo simulator (Todorov et al., 2012) and provided in the OpenAI Gym (Brockman et al., 2016) benchmark for online RL. We include the HalfCheetah, Hopper, Walker2D, and Ant tasks. We include this domain primarily for comparison with past works, as a vast array of popular RL

| Statistics | cartpole swingup | cheetah run | finger turn hard | fish swim | humanoid run | walker stand | walker walk | manipulator insert ball | manipulator insert peg |
|---|---|---|---|---|---|---|---|---|---|
| Dataset size | 40K | 300K | 500K | 200K | 3M | 200K | 200K | 1.5M | 1.5M |
| State dim. | 5 | 17 | 12 | 24 | 67 | 24 | 24 | 44 | 44 |
| Action dim. | 1 | 6 | 2 | 5 | 21 | 6 | 6 | 5 | 5 |
| Properties | - | H, L | H, L | H, L | H, L | H, L | H, L | H, L, T | H, L,T |

| Statistics | maze2d | antmaze | halfcheetah | hopper | walker | ant | hammer | door | relocate | pen |
|---|---|---|---|---|---|---|---|---|---|---|
| Dataset size | 1/2/4M | 1M | 1M | 1M | 1M | 1M | 11K/1M | 7K/1M | 10K/1M | 5K/500K |
| # datasets | 1 | 1 | 5 | 5 | 5 | 5 | 3 | 3 | 3 | 3 |
| State dim. | 4 | 29 | 17 | 11 | 17 | 111 | 46 | 39 | 39 | 45 |
| Action dim. | 2 | 8 | 6 | 3 | 6 | 8 | 26 | 28 | 30 | 24 |
| Properties | T | T, R | H | H | H | H | H, R | H, R | H, R | H, R |

Table 1: Task statistics for RLUnplugged tasks (top) and D4RL tasks (bottom). Dataset size is the number of $(s, a, r, s')$ tuples. For each dataset, we note the properties it possesses: high dimensional spaces (**H**), long time-horizon (**L**), sparse rewards (**R**), temporally extended control (**T**).

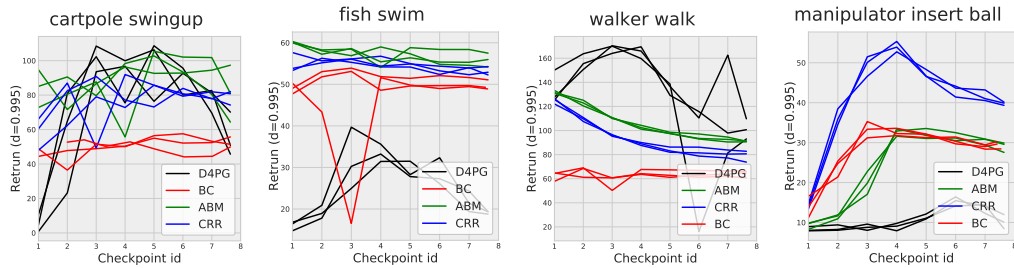

Figure 3: Online evaluation of policy checkpoints for 4 Offline RL algorithms with 3 random seeds. We observe a large degree of variability between the behavior of algorithms on different tasks. Without online evaluation, tuning the hyperparameters (e.g., choice of Offline RL algorithm and policy checkpoint) is challenging. This highlights the practical importance of Offline policy selection when online evaluation is not feasible. See Figure A.7 for additional tasks.

methods have been evaluated and developed on these tasks (Schulman et al., 2015; Lillicrap et al., 2015; Schulman et al., 2017; Fujimoto et al., 2018; Haarnoja et al., 2018).

**Gym-MuJoCo datasets and policies.** For each task, in order to explore the effect of varying distributions, we include 5 datasets originally proposed by Fu et al. (2020). 3 correspond to different performance levels of the agent – 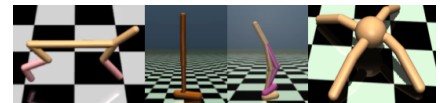
"random", "medium", and "expert". We additionally include a mixture of medium and expert dataset, labeled "medium-expert", and data collected from a replay buffer until the policy reaches the medium level of performance, labeled "medium-replay". For policies, we selected 11 policies collected from evenly-spaced snapshots of training a Soft Actor-Critic agent (Haarnoja et al., 2018), which covers a range of performance between random and expert.

**Maze2D and AntMaze tasks.** Maze2D and AntMaze are two maze navigation tasks originally proposed in D4RL (Fu et al., 2020). The domain consists of 3 mazes ranging from easy to hard ("umaze", "medium", "large"), and two morphologies: a 2D ball in Maze2D and the "Ant" robot of the Gym benchmark in AntMaze. For Maze2D, 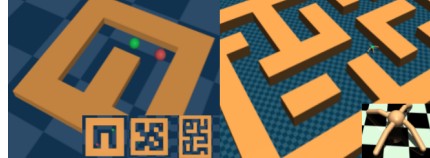
we provide a less challenging reward computed base on distance to a fixed goal. For the AntMaze environment reward is given only upon reaching the fixed goal.

**Maze2D and AntMaze datasets and policies.** Datasets for both morphologies consists of undirect data navigating randomly to different goal locations. The datasets for Maze2D are collected by using a high-level planner to command waypoints to a low-level PID controller in order to reach randomly selected goals. The dataset in AntMaze is generated using the same high-level planner, but the low-

level planner is replaced with a goal-conditioned policy trained to reach arbitrary waypoints. Both of these datasets are generated from non-Markovian policies, as the high-level controller maintains a history of waypoints reached in order to construct a plan to the goal. We provide policies for all environments except "antmaze-large" by taking training snapshots obtained while running the DAPG algorithm (Rajeswaran et al., 2017). Because obtaining high-performing policies for "antmaze-large" was challenging, we instead used imitation learning on a large amount of expert data to generate evaluation policies. This expert data is obtained by collecting additional trajectories that reach the goal using a high-level waypoint planner in conjunction with a low-level goal-conditioned policy (this is the same method as was used to generate the dataset, Sec. 5 (Fu et al., 2020)).

**Adroit tasks.** The Adroit domain is a realistic simulation based on the Shadow Hand robot, first proposed by Rajeswaran et al. (2017). There are 4 tasks in this domain: opening a door ("door"), pen twirling ("pen"), moving a ball to a target location ("relocate"), and hitting a nail with a hammer ("hammer"). These tasks all contain sparse rewards and are difficult to learn without demonstrations.

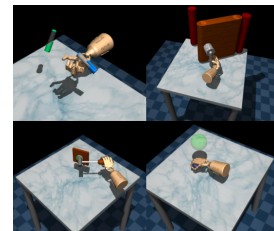

**Adroit datasets and policies.** We include 3 datasets for each task. The "human" dataset consists of a small amount of human demonstrations performing the task. The "expert" dataset consists of data collected from an expert trained via DAPG (Rajeswaran et al., 2017). Finally, the "cloned" dataset contains a mixture of human demonstrations and data collected from an imitation learning algorithm trained on the demonstrations. For policies, we include 11 policies collected from snapshots while running the DAPG algorithm, which range from random performance to expert performance.

## 5 BASELINES AND RESULTS

The goal of our evaluation is two-fold. First, we wish to measure the performance of a variety of existing algorithms to provide baselines and reference numbers for future research. Second, we wish to identify shortcomings in these approaches to reveal promising directions for future research.

### 5.1 BASELINES

We selected six methods to evaluate, which cover a variety of approaches that have been explored for the OPE problem.

**Fitted Q-Evaluation (FQE)** As in Le et al. (2019), we train a neural network to estimate the value of the evaluation policy $\pi$ by bootstrapping from $Q(s', \pi(s'))$. We tried two different implementations, one from Kostrikov & Nachum (2020)[3] and another from Paine et al. (2020) labeled FQE-L2 and FQE-D respectively to reflect different choices in loss function and parameterization.

**Model-Based (MB)** Similar to Paduraru (2007), we train dynamics and reward models on transitions from the offline dataset $\mathcal{D}$. Our models are deep neural networks trained to maximize the log likelihood of the next state and reward given the current state and action, similar to models from successful model-based RL algorithms (Chua et al., 2018; Janner et al., 2019). We follow the setup detailed in Zhang et al. (2021). We include both the feed-forward and auto-regressive models labeled MB-FF and MB-AR respectively. To evaluate a policy, we compute the return using simulated trajectories generated by the policy under the learned dynamics model.

**Importance Sampling (IS)** We perform importance sampling with a learned behavior policy. We use the implementation from Kostrikov & Nachum (2020)[3], which uses self-normalized (also known as weighted) step-wise importance sampling (Precup, 2000). Since the behavior policy is not known explicitly, we learn an estimate of it via a max-likelihood objective over the dataset $\mathcal{D}$, as advocated by Xie et al. (2018); Hanna et al. (2019). In order to be able to compute log-probabilities when the target policy is deterministic, we add artificial Gaussian noise with standard deviation 0.01 for all deterministic target policies.

---

[3]Code available at `https://github.com/google-research/google-research/tree/master/policy_eval`.

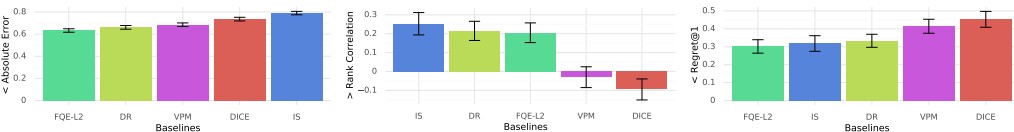

Figure 4: **DOPE RL Unplugged** Mean overall performance of baselines.

Figure 5: **DOPE D4RL** Mean overall performance of baselines.

**Doubly-Robust (DR)** We perform weighted doubly-robust policy evaluation Thomas & Brunskill (2016) using the implementation of Kostrikov & Nachum (2020)[3]. Specifically, this method combines the IS technique above with a value estimator for variance reduction. The value estimator is learned using deep FQE with an L2 loss function. More advanced approaches that trade variance for bias exist (e.g., MAGIC (Thomas & Brunskill, 2016)), but we leave implementing them to future work.

**DICE** This method uses a saddle-point objective to estimate marginalized importance weights $d^{\pi}(s,a)/d^{\pi_B}(s,a)$; these weights are then used to compute a weighted average of reward over the offline dataset, and this serves as an estimate of the policy's value in the MDP. We use the implementation from Yang et al. (2020) corresponding to the algorithm *BestDICE*.[4]

**Variational Power Method (VPM)** This method runs a variational power iteration algorithm to estimate the importance weights $d^{\pi}(s,a)/d^{\pi_B}(s,a)$ without the knowledge of the behavior policy. It then estimates the target policy value using weighted average of rewards similar to the DICE method. Our implementation is based on the same network and hyperparameters for OPE setting as in Wen et al. (2020). We further tune the hyper-parameters including the regularization parameter $\lambda$, learning rates $\alpha_{\theta}$ and $\alpha_v$, and number of iterations on the Cartpole swingup task using ground-truth policy value, and then fix them for all other tasks.

## 5.2 RESULTS

To facilitate aggregate metrics and comparisons between tasks and between DOPE RL Unplugged and DOPE D4RL, we normalize the returns and estimated returns to range between 0 and 1. For each set of policies we compute the worst value $V_{worst} = min\{V^{\pi_1}, V^{\pi_2}, ..., V^{\pi_N}\}$ and best value $V_{best} = max\{V^{\pi_1}, V^{\pi_2}, ..., V^{\pi_N}\}$ and normalize the returns and estimated returns according to $x' = (x - V_{worst})/(V_{best} - V_{worst})$.

We present results averaged across DOPE RL Unplugged in Fig. 4, and results for DOPE D4RL in Fig. 5. Overall, no evaluated algorithm attains near-oracle performance under any metric (absolute error, regret, or rank correlation). Because the dataset is finite, we do not expect that achieving oracle performance is possible. Nevertheless, based on recent progress on this benchmark (e.g., Zhang et al. (2021)), we hypothesize that the benchmark has room for improvement, making it suitable for driving further improvements on OPE methods and facilitating the development of OPE algorithms that can provide reliable estimates on the types of high-dimensional problems that we consider.

While all algorithms achieve sub-optimal performance, some perform better than others. We find that on the DOPE RL Unplugged tasks model based (MB-AR, MB-FF) and direct value based methods (FQE-D, FQE-L2) significantly outperform importance sampling methods (VPM, DICE, IS) across all metrics. This is somewhat surprising as DICE and VPM have shown promising results in other settings. We hypothesize that this is due to the relationship between the behavior data and evaluation policies, which is different from standard OPE settings. Recall that in DOPE RL Unplugged the behavior data is collected from an online RL algorithm and the evaluation policies are learned via offline RL from the behavior data. In our experience all methods work better when the behavior policy is a noisy/perturbed version of the evaluation policy. Moreover, MB and FQE-based methods may

---

[4]Code available at `https://github.com/google-research/dice_rl`.

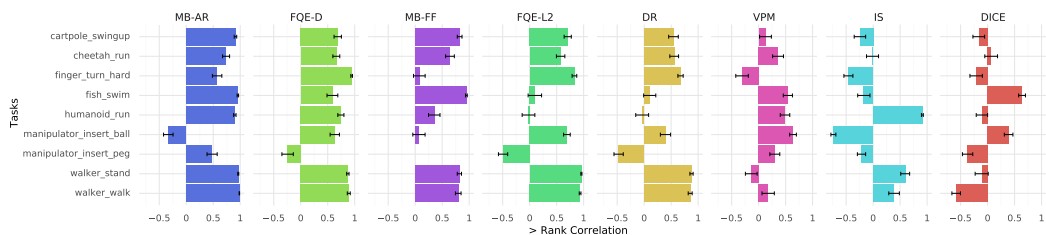

Figure 6: Rank correlation for each baseline algorithm for each RL Unplugged task considered.

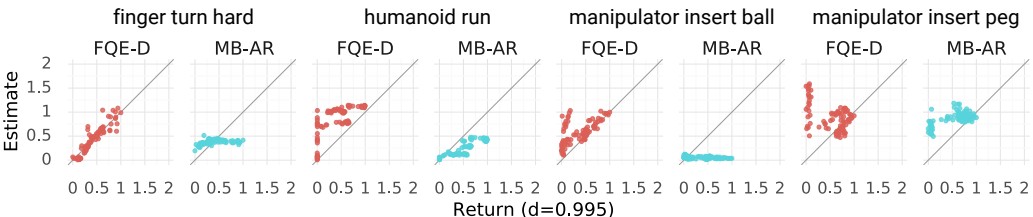

Figure 7: Scatter plots of estimate vs ground truth return for MB-AR and FQE-D on selected tasks.

implicitly benefit from the architectural and optimization advancements made in policy optimization settings, which focus on similar environments and where these methods are more popular than importance sampling approaches. Note that within the MB and FQE methods, design details can create a significant difference in performance. For example model architecture (MB-AR vs MB-FF) and implementation differences (FQE-D vs FQE-L2) show differing performance on certain tasks.

On DOPE D4RL, direct value based methods still do well, with FQE-L2 performing best on the Absolute Error and Regret@1 metrics. However, there are cases where other methods outperform FQE. Notably, IS and DR outperform FQE-L2 under the rank correlation metric. As expected, there is a clear performance gap between DOPE RL Unplugged and DOPE D4RL. While both domains have challenging tasks, algorithms perform better under the more ideal conditions of DOPE RL Unplugged than under the challenging conditions of DOPE D4RL (0.69 vs 0.25 rank correlation respectively).

In Fig. A.2 we show the rank correlation for each task in DOPE RL Unplugged. Most tasks follow the overall trends, but we will highlight a few exceptions. 1) Importance sampling is among the best methods for the *humanoid run* task, significantly outperforming direct value-based methods. 2) while MB-AR and FQE-D are similar overall, there are a few tasks where the difference is large, for example FQE-D outperfroms MB-AR on *finger turn hard*, and *manipulator insert ball*, where as MB-AR outperforms FQE-D on *cartpole swingup*, *fish swim*, *humanoid run*, and *manipulator insert peg*. We show the scatter plots for MB-AR and FQE-D on these tasks in Fig 7 which highlights different failure modes: when MB-AR performs worse, it assigns similar values for all policies; when FQE-D performs worse, it severely over-estimates the values of poor policies.

We present more detailed results, separated by task, in Appendix A.2. Note in particular how in Table A.2.2, which shows the regret@1 metric for different D4RL tasks, the particular choice of dataset for the Gym-MuJoCo, Adroit, and AntMaze domains causes a significant difference in the performance of OPE methods. This indicates the importance of evaluating multiple distinct datasets, with different data distribution properties (e.g., more narrow datasets, such as expert data, vs. broader datasets, such as random data), as no tested method is reliably robust to the effects of dataset variation.

High-dimensional tasks requiring temporally extended control were also challenging, as highlighted by the performance on the AntMaze domain. No algorithm was able to achieve a good absolute error value on such tasks, and importance sampling was the only method able to achieve a correlation consistently above zero, suggesting that these more complex tasks are a particularly important area for future methods to focus on.

## 6 RELATED WORK

Off-policy evaluation (OPE) has been studied extensively across a range of different domains, from healthcare (Thapa et al., 2005; Raghu et al., 2018; Nie et al., 2019), to recommender systems (Li et al., 2010; Dudík et al., 2014; Theocharous et al., 2015), and robotics (Kalashnikov et al., 2018). While a full survey of OPE methods is outside the scope of this article, broadly speaking we can categories OPE methods into groups based the use of importance sampling (Precup, 2000), value functions (Sutton et al., 2009; Migliavacca et al., 2010; Sutton et al., 2016; Yang et al., 2020), and learned transition models (Paduraru, 2007), though a number of methods combine two or more of these components (Jiang & Li, 2015; Thomas & Brunskill, 2016; Munos et al., 2016). A significant body of work in OPE is also concerned with providing statistical guarantees (Thomas et al., 2015). Our focus instead is on empirical evaluation – while theoretical analysis is likely to be a critical part of future OPE research, combining such analysis with empirical demonstration on broadly accepted and standardized benchmarks is likely to facilitate progress toward practically useful algorithms.

Current evaluation of OPE methods is based around several metrics, including error in predicting the true return of the evaluated policy (Voloshin et al., 2019), correlation between the evaluation output and actual returns (Irpan et al., 2019), and ranking and model selection metrics (Doroudi et al., 2017). As there is no single accepted metric used by the entire community, we provide a set of candidate metrics along with our benchmark, with a detailed justification in Section 5. Our work is closely related to (Paine et al., 2020) which studies OPE in a similar setting, however in our work we present a benchmark for the community and compare a range of OPE methods. Outside of OPE, standardized benchmark suites have led to considerable standardization and progress in RL (Stone & Sutton, 2001; Dutech et al., 2005; Riedmiller et al., 2007). The Arcade Learning Environment (ALE) (Bellemare et al., 2013) and OpenAI Gym (Brockman et al., 2016) have been widely used to compare online RL algorithms to good effect. More recently, Gulcehre et al. (2020); Fu et al. (2020) proposed benchmark tasks for offline RL. Our benchmark is based on the tasks and environments described in these two benchmarks, which we augment with a set of standardized policies for evaluation, results for a number of existing OPE methods, and standardized evaluation metrics and protocols. Voloshin et al. (2019) have recently proposed benchmarking for OPE methods on a variety of tasks ranging from tabular problems to image-based tasks in Atari. Our work differs in several key aspects. Voloshin et al. (2019) is composed entirely of discrete action tasks, whereas out benchmark focuses on continuous action tasks. Voloshin et al. (2019) assumes full support for the evaluation policy under the behavior policy data, whereas we designed our datasets and policies to ensure that different cases of dataset and policy distributions could be studied. Finally, all evaluations in Voloshin et al. (2019) are performed using the MSE metric, and they do not provide standardized datasets. In contrast, we provide a variety of policies for each problem which enables one to evaluate metrics such as ranking for policy selection, and a wide range of standardized datasets for reproducbility.

## 7 CONCLUSION

We have presented the Deep Off-Policy Evaluation (DOPE) benchmark, which aims to provide a platform for studying policy evaluation and selection across a wide range of challenging tasks and datasets. In contrast to prior benchmarks, DOPE provides multiple datasets and policies, allowing researchers to study how data distributions affect performance and to evaluate a wide variety of metrics, including those that are relevant for offline policy selection. In comparing existing OPE methods, we find that no existing algorithms consistently perform well across all of the tasks, which further reinforces the importance of standardized and challenging OPE benchmarks. Moreover, algorithms that perform poorly under one metric, such as absolute error, may perform better on other metrics, such as correlation, which provides insight into what algorithms to use depending on the use case (e.g., policy evaluation vs. policy selection).

We believe that OPE is an exciting area for future research, as it allows RL agents to learn from large and abundant datasets in domains where online RL methods are otherwise infeasible. We hope that our benchmark will enable further progress in this field, though important evaluation challenges remain. As the key benefit of OPE is the ability to utilize real-world datasets, a promising direction for future evaluation efforts is to devise effective ways to use such data, where a key challenge is to develop evaluation protocols that are both reproducible and accessible. This could help pave the way towards developing intelligent decision making agents that can leverage vast banks of logged information to solve important real-world problems.

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

# A APPENDIX

## A.1 METRICS

The metrics we use in our paper are defined as follows:

**Absolute Error** We evaluate policies using absolute error in order to be robust to outliers. The absolute error is defined as the difference between the value and estimated value of a policy:

$$\text{AbsErr} = |V^\pi - \hat{V}^\pi| \tag{2}$$

Where $V^\pi$ is the true value of the policy, and $\hat{V}^\pi$ is the estimated value of the policy.

**Regret@k** Regret@k is the difference between the value of the best policy in the entire set, and the value of the best policy in the top-k set (where the top-k set is chosen by estimated values). It can be defined as:

$$\text{Regret @ k} = \max_{i \in 1:N} V_i^\pi - \max_{j \in \text{topk}(1:N)} V_j^\pi \tag{3}$$

Where $\text{topk}(1:N)$ denotes the indices of the top K policies as measured by estimated values $\hat{V}^\pi$.

**Rank correlation** Rank correlation (also Spearman's $\rho$) measures the correlation between the ordinal rankings of the value estimates and the true values. It can be written as:

$$\text{RankCorr} = \frac{\text{Cov}(V_{1:N}^\pi, \hat{V}_{1:N}^\pi)}{\sigma(V_{1:N}^\pi)\sigma(\hat{V}_{1:N}^\pi)} \tag{4}$$

## A.2 DETAILED RESULTS

Detailed results figures and tables are presented here. We show results by task in both tabular and chart form, as well as scatter plots which compare the estimated returns against the ground truth returns for every policy.

### A.2.1 CHART RESULTS

First we show the normalized results for each algorithm and task.

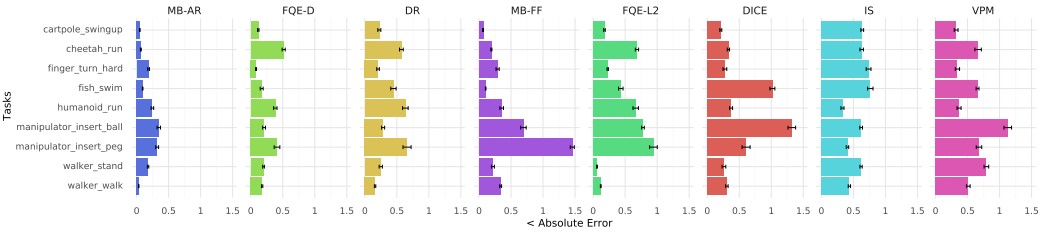

Figure A.1: Absolute error for each baseline algorithm for each RL Unplugged task considered.

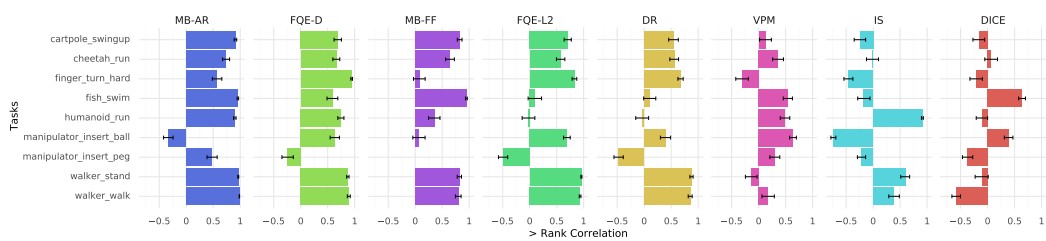

Figure A.2: Rank correlation for each baseline algorithm for each RL Unplugged task considered.

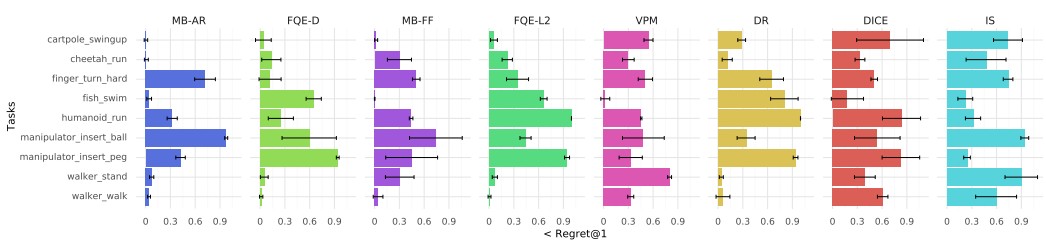

Figure A.3: Regret@1 for each baseline algorithm for each RL Unplugged task considered.

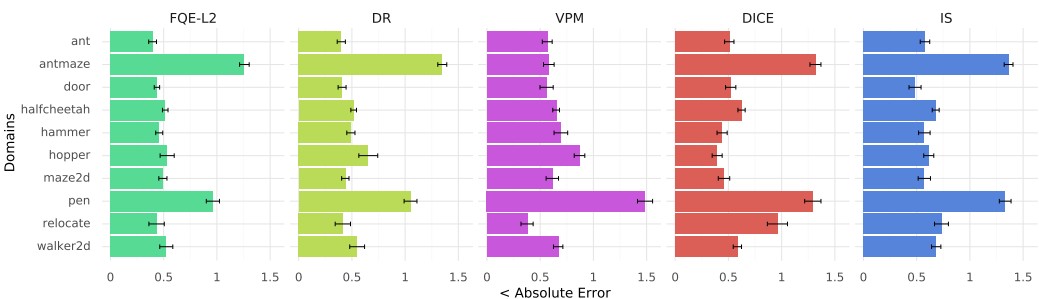

Figure A.4: Absolute error for each baseline algorithm for each D4RL task domain considered.

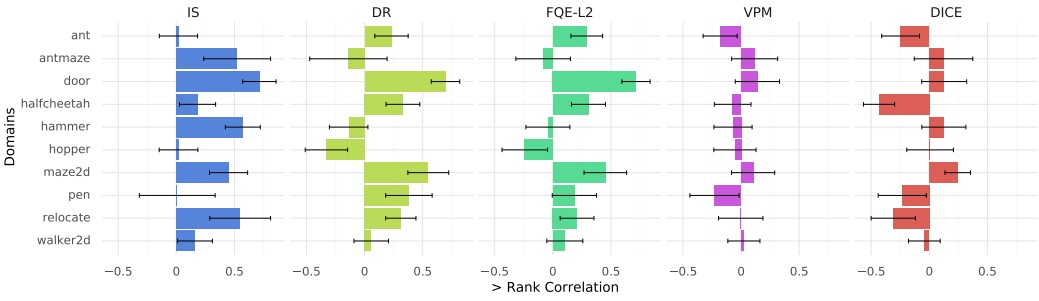

Figure A.5: Rank correlation for each baseline algorithm for each D4RL task domain considered.

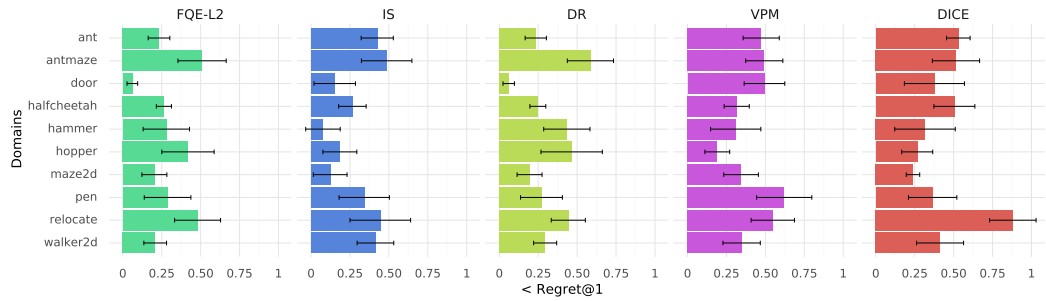

Figure A.6: Regret@1 for each baseline algorithm for each D4RL task domain considered.

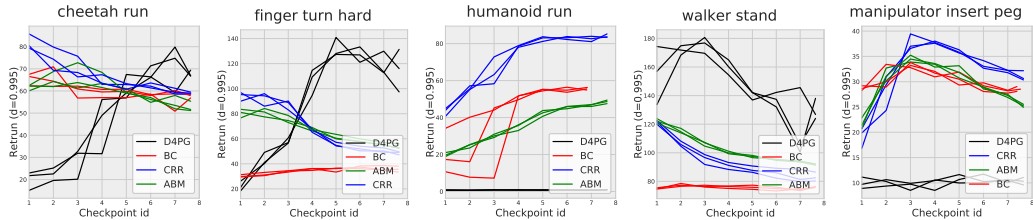

Figure A.7: Online evaluation of policy checkpoints for 4 Offline RL algorithms with 3 random seeds. We observe a large degree of variability between the behavior of algorithms on different tasks.

### A.2.2   TABULAR RESULTS

Next, we present the results for each task and algorithm in tabular form, with means and standard deviations reported across 3 seeds.

| | | Cartpole swingup | Cheetah run | Finger turn hard | Fish swim | Humanoid run |
|---|---|---|---|---|---|---|
| **Absolute Error** btw. OPE and ground truth | Variational power method | $37.53_{\pm3.50}$ | $61.89_{\pm4.25}$ | $46.22_{\pm3.93}$ | $31.27_{\pm0.99}$ | $35.29_{\pm3.03}$ |
| | Importance Sampling | $68.75_{\pm2.39}$ | $44.29_{\pm1.91}$ | $90.10_{\pm4.68}$ | $34.82_{\pm1.93}$ | $27.89_{\pm1.98}$ |
| | Best DICE | $22.73_{\pm1.65}$ | $23.35_{\pm1.32}$ | $33.52_{\pm3.48}$ | $59.48_{\pm2.47}$ | $31.42_{\pm2.04}$ |
| | Model based - FF | $\mathbf{6.80}_{\pm\mathbf{0.85}}$ | $13.64_{\pm0.59}$ | $35.99_{\pm3.00}$ | $\mathbf{4.75}_{\pm\mathbf{0.23}}$ | $30.12_{\pm2.40}$ |
| | FQE (L2) | $19.02_{\pm1.34}$ | $48.26_{\pm1.78}$ | $27.91_{\pm1.18}$ | $19.82_{\pm1.57}$ | $56.28_{\pm3.52}$ |
| | Doubly Robust (IS, FQE) | $24.38_{\pm2.51}$ | $40.27_{\pm2.05}$ | $25.26_{\pm2.48}$ | $20.28_{\pm1.90}$ | $53.64_{\pm3.68}$ |
| | FQE (distributional) | $12.63_{\pm1.21}$ | $36.50_{\pm1.62}$ | $\mathbf{10.23}_{\pm\mathbf{0.93}}$ | $7.76_{\pm0.95}$ | $32.36_{\pm2.27}$ |
| | Model based - AR | $\mathbf{5.32}_{\pm\mathbf{0.54}}$ | $\mathbf{4.64}_{\pm\mathbf{0.46}}$ | $22.93_{\pm1.72}$ | $\mathbf{4.31}_{\pm\mathbf{0.22}}$ | $\mathbf{20.95}_{\pm\mathbf{1.61}}$ |

| | | Walker stand | Walker walk | Manipulator insert ball | Manipulator insert peg | Median ↓ |
|---|---|---|---|---|---|---|
| **Absolute Error** btw. OPE and ground truth | Variational power method | $96.76_{\pm3.59}$ | $87.24_{\pm4.25}$ | $79.25_{\pm6.19}$ | $21.95_{\pm1.17}$ | 46.22 |
| | Importance Sampling | $66.50_{\pm1.90}$ | $67.24_{\pm2.70}$ | $29.93_{\pm1.10}$ | $12.78_{\pm0.66}$ | 44.29 |
| | Best DICE | $27.58_{\pm3.01}$ | $47.28_{\pm3.13}$ | $103.45_{\pm5.21}$ | $22.75_{\pm3.00}$ | 31.42 |
| | Model based - FF | $23.34_{\pm2.41}$ | $52.23_{\pm2.34}$ | $34.30_{\pm2.55}$ | $121.12_{\pm1.58}$ | 30.12 |
| | FQE (L2) | $\mathbf{6.51}_{\pm\mathbf{0.71}}$ | $18.34_{\pm0.95}$ | $36.32_{\pm1.07}$ | $31.12_{\pm2.37}$ | 27.91 |
| | Doubly Robust (IS, FQE) | $26.82_{\pm2.66}$ | $24.63_{\pm1.69}$ | $13.33_{\pm1.16}$ | $22.28_{\pm2.34}$ | 24.63 |
| | FQE (distributional) | $21.49_{\pm1.41}$ | $27.57_{\pm1.54}$ | $\mathbf{9.75}_{\pm\mathbf{1.10}}$ | $12.66_{\pm1.39}$ | 12.66 |
| | Model based - AR | $19.12_{\pm1.23}$ | $\mathbf{5.14}_{\pm\mathbf{0.49}}$ | $17.13_{\pm1.34}$ | $\mathbf{9.71}_{\pm\mathbf{0.70}}$ | 9.71 |

Table A.1:  Average absolute error between OPE metrics and ground truth values at a discount factor of 0.995 In each column, absolute error values that are not significantly different from the best ($p > 0.05$) are bold faced. Methods are ordered by median.

| | | Cartpole swingup | Cheetah run | Finger turn hard | Fish swim | Humanoid run |
|---|---|---|---|---|---|---|
| Rank Correlation btw. OPE and ground truth | Importance Sampling | $-0.23_{\pm0.11}$ | $-0.01_{\pm0.12}$ | $-0.45_{\pm0.08}$ | $-0.17_{\pm0.11}$ | $\mathbf{0.91}_{\pm\mathbf{0.02}}$ |
| | Best DICE | $-0.16_{\pm0.11}$ | $0.07_{\pm0.11}$ | $-0.22_{\pm0.11}$ | $0.44_{\pm0.09}$ | $-0.10_{\pm0.10}$ |
| | Variational power method | $0.01_{\pm0.11}$ | $0.01_{\pm0.12}$ | $-0.25_{\pm0.11}$ | $0.56_{\pm0.08}$ | $0.36_{\pm0.09}$ |
| | Doubly Robust (IS, FQE) | $0.55_{\pm0.09}$ | $0.56_{\pm0.08}$ | $0.67_{\pm0.05}$ | $0.11_{\pm0.12}$ | $-0.03_{\pm0.12}$ |
| | Model based - FF | $0.83_{\pm0.05}$ | $\mathbf{0.64}_{\pm\mathbf{0.08}}$ | $0.08_{\pm0.11}$ | $\mathbf{0.95}_{\pm\mathbf{0.02}}$ | $0.35_{\pm0.10}$ |
| | FQE (distributional) | $0.69_{\pm0.07}$ | $\mathbf{0.67}_{\pm\mathbf{0.06}}$ | $\mathbf{0.94}_{\pm\mathbf{0.01}}$ | $0.59_{\pm0.10}$ | $0.74_{\pm0.06}$ |
| | FQE (L2) | $0.70_{\pm0.07}$ | $0.56_{\pm0.08}$ | $0.83_{\pm0.04}$ | $0.10_{\pm0.12}$ | $-0.02_{\pm0.12}$ |
| | Model based - AR | $\mathbf{0.91}_{\pm\mathbf{0.02}}$ | $\mathbf{0.74}_{\pm\mathbf{0.07}}$ | $0.57_{\pm0.09}$ | $\mathbf{0.96}_{\pm\mathbf{0.01}}$ | $\mathbf{0.90}_{\pm\mathbf{0.02}}$ |

| | | Walker stand | Walker walk | Manipulator insert ball | Manipulator insert peg | Median ↑ |
|---|---|---|---|---|---|---|
| Rank Correlation btw. OPE and ground truth | Importance Sampling | $0.59_{\pm0.08}$ | $0.38_{\pm0.10}$ | $-0.72_{\pm0.05}$ | $-0.25_{\pm0.08}$ | $-0.17$ |
| | Best DICE | $-0.11_{\pm0.12}$ | $-0.58_{\pm0.08}$ | $0.19_{\pm0.11}$ | $-0.35_{\pm0.10}$ | $-0.11$ |
| | Variational power method | $-0.35_{\pm0.10}$ | $-0.10_{\pm0.11}$ | $\mathbf{0.61}_{\pm\mathbf{0.08}}$ | $\mathbf{0.41}_{\pm\mathbf{0.09}}$ | $0.01$ |
| | Doubly Robust (IS, FQE) | $0.88_{\pm0.03}$ | $0.85_{\pm0.04}$ | $0.42_{\pm0.10}$ | $-0.47_{\pm0.09}$ | $0.55$ |
| | Model based - FF | $0.82_{\pm0.04}$ | $0.80_{\pm0.05}$ | $0.06_{\pm0.10}$ | $-0.56_{\pm0.08}$ | $0.64$ |
| | FQE (distributional) | $0.87_{\pm0.02}$ | $0.89_{\pm0.03}$ | $\mathbf{0.63}_{\pm\mathbf{0.08}}$ | $-0.23_{\pm0.10}$ | $0.69$ |
| | FQE (L2) | $\mathbf{0.96}_{\pm\mathbf{0.01}}$ | $0.94_{\pm0.02}$ | $\mathbf{0.70}_{\pm\mathbf{0.07}}$ | $-0.48_{\pm0.08}$ | $0.70$ |
| | Model Based - AR | $\mathbf{0.96}_{\pm\mathbf{0.01}}$ | $\mathbf{0.98}_{\pm\mathbf{0.00}}$ | $-0.33_{\pm0.09}$ | $\mathbf{0.47}_{\pm\mathbf{0.09}}$ | $0.90$ |

Table A.2: Spearman's rank correlation ($\rho$) coefficient (bootstrap mean $\pm$ standard deviation) between different OPE metrics and ground truth values at a discount factor of 0.995. In each column, rank correlation coefficients that are not significantly different from the best ($p > 0.05$) are bold faced. Methods are ordered by median. Also see Table A.3 and Table A.1 for Normalized Regret@5 and Average Absolute Error results.

| | | Cartpole swingup | Cheetah run | Finger turn hard | Fish swim | Humanoid run |
|---|---|---|---|---|---|---|
| Regret@5 for OPE vs. ground truth | Importance Sampling | $0.73_{\pm0.16}$ | $0.40_{\pm0.21}$ | $0.64_{\pm0.05}$ | $0.12_{\pm0.05}$ | $\mathbf{0.31}_{\pm\mathbf{0.09}}$ |
| | Best DICE | $0.68_{\pm0.41}$ | $0.27_{\pm0.05}$ | $0.44_{\pm0.04}$ | $\mathbf{0.35}_{\pm\mathbf{0.24}}$ | $0.84_{\pm0.22}$ |
| | Variational power method | $0.50_{\pm0.13}$ | $0.37_{\pm0.04}$ | $0.45_{\pm0.13}$ | $\mathbf{0.02}_{\pm\mathbf{0.02}}$ | $0.56_{\pm0.08}$ |
| | Doubly Robust (IS, FQE) | $0.28_{\pm0.05}$ | $\mathbf{0.09}_{\pm\mathbf{0.05}}$ | $0.56_{\pm0.12}$ | $0.61_{\pm0.12}$ | $0.99_{\pm0.00}$ |
| | FQE (L2) | $0.06_{\pm0.04}$ | $0.17_{\pm0.05}$ | $\mathbf{0.30}_{\pm\mathbf{0.11}}$ | $0.50_{\pm0.03}$ | $0.99_{\pm0.00}$ |
| | Model based - FF | $\mathbf{0.02}_{\pm\mathbf{0.02}}$ | $0.24_{\pm0.12}$ | $0.43_{\pm0.04}$ | $\mathbf{0.00}_{\pm\mathbf{0.00}}$ | $0.44_{\pm0.02}$ |
| | FQE (distributional) | $\mathbf{0.03}_{\pm\mathbf{0.09}}$ | $\mathbf{0.11}_{\pm\mathbf{0.09}}$ | $\mathbf{0.10}_{\pm\mathbf{0.12}}$ | $0.49_{\pm0.06}$ | $\mathbf{0.24}_{\pm\mathbf{0.15}}$ |
| | Model based - AR | $\mathbf{0.00}_{\pm\mathbf{0.02}}$ | $\mathbf{0.01}_{\pm\mathbf{0.02}}$ | $0.63_{\pm0.11}$ | $\mathbf{0.03}_{\pm\mathbf{0.02}}$ | $\mathbf{0.32}_{\pm\mathbf{0.06}}$ |

| | | Walker stand | Walker walk | Manipulator insert ball | Manipulator insert peg | Median ↓ |
|---|---|---|---|---|---|---|
| Regret@5 for OPE vs. ground truth | Importance Sampling | $0.54_{\pm0.11}$ | $0.54_{\pm0.23}$ | $0.83_{\pm0.05}$ | $\mathbf{0.22}_{\pm\mathbf{0.03}}$ | $0.54$ |
| | Best DICE | $0.24_{\pm0.07}$ | $0.55_{\pm0.06}$ | $\mathbf{0.44}_{\pm\mathbf{0.07}}$ | $0.75_{\pm0.04}$ | $0.44$ |
| | Variational power method | $0.41_{\pm0.02}$ | $0.39_{\pm0.02}$ | $\mathbf{0.52}_{\pm\mathbf{0.20}}$ | $0.32_{\pm0.02}$ | $0.41$ |
| | Doubly Robust (IS, FQE) | $\mathbf{0.02}_{\pm\mathbf{0.01}}$ | $\mathbf{0.05}_{\pm\mathbf{0.07}}$ | $0.30_{\pm0.10}$ | $0.73_{\pm0.01}$ | $0.30$ |
| | FQE (L2) | $\mathbf{0.04}_{\pm\mathbf{0.02}}$ | $\mathbf{0.00}_{\pm\mathbf{0.02}}$ | $\mathbf{0.37}_{\pm\mathbf{0.07}}$ | $0.74_{\pm0.01}$ | $0.30$ |
| | Model based - FF | $0.18_{\pm0.10}$ | $\mathbf{0.03}_{\pm\mathbf{0.05}}$ | $0.83_{\pm0.06}$ | $0.74_{\pm0.01}$ | $0.24$ |
| | FQE (distributional) | $\mathbf{0.03}_{\pm\mathbf{0.03}}$ | $\mathbf{0.01}_{\pm\mathbf{0.02}}$ | $0.50_{\pm0.30}$ | $0.73_{\pm0.01}$ | $0.11$ |
| | Model based - AR | $\mathbf{0.04}_{\pm\mathbf{0.02}}$ | $\mathbf{0.04}_{\pm\mathbf{0.02}}$ | $0.85_{\pm0.02}$ | $\mathbf{0.30}_{\pm\mathbf{0.04}}$ | $0.04$ |

Table A.3: Normalized Regret@5 (bootstrap mean $\pm$ standard deviation) for OPE methods *vs.* ground truth values at a discount factor of 0.995. In each column, normalized regret values that are not significantly different from the best ($p > 0.05$) are bold faced. Methods are ordered by median.

| | | Halfcheetah expert | Halfcheetah medium | Halfcheetah medium-expert | Halfcheetah medium-replay | Halfcheetah random |
|---|---|---|---|---|---|---|
| Abs. Error | IS | $1404_{\pm152}$ | $\mathbf{1217}_{\pm123}$ | $1400_{\pm146}$ | $1409_{\pm154}$ | $1405_{\pm155}$ |
| | VPM | $\mathbf{945}_{\pm164}$ | $\mathbf{1374}_{\pm153}$ | $1427_{\pm111}$ | $1384_{\pm148}$ | $1411_{\pm154}$ |
| | Best DICE | $\mathbf{944}_{\pm161}$ | $\mathbf{1382}_{\pm130}$ | $\mathbf{1078}_{\pm132}$ | $1440_{\pm158}$ | $1446_{\pm156}$ |
| | Doubly Robust | $\mathbf{1025}_{\pm95}$ | $\mathbf{1222}_{\pm134}$ | $\mathbf{1015}_{\pm103}$ | $\mathbf{1001}_{\pm129}$ | $\mathbf{949}_{\pm126}$ |
| | FQE (L2) | $\mathbf{1031}_{\pm95}$ | $\mathbf{1211}_{\pm130}$ | $\mathbf{1014}_{\pm101}$ | $\mathbf{1003}_{\pm132}$ | $\mathbf{938}_{\pm125}$ |

| | | Antmaze large-diverse | Antmaze large-play | Antmaze medium-diverse | Antmaze medium-play | Antmaze umaze |
|---|---|---|---|---|---|---|
| Abs. Error | IS | $0.62_{\pm0.01}$ | $0.85_{\pm0.00}$ | $0.55_{\pm0.01}$ | $0.81_{\pm0.00}$ | $0.62_{\pm0.04}$ |
| | VPM | $\mathbf{0.02}_{\pm0.02}$ | $\mathbf{0.26}_{\pm0.24}$ | $\mathbf{0.07}_{\pm0.05}$ | $\mathbf{0.11}_{\pm0.06}$ | $\mathbf{0.12}_{\pm0.03}$ |
| | Best DICE | $5.55_{\pm0.36}$ | $19.62_{\pm1.28}$ | $2.42_{\pm1.56}$ | $19.47_{\pm2.15}$ | $14.97_{\pm1.93}$ |
| | Doubly Robust | $0.99_{\pm0.01}$ | $1.59_{\pm0.01}$ | $0.61_{\pm0.03}$ | $1.47_{\pm0.01}$ | $0.87_{\pm0.04}$ |
| | FQE (L2) | $0.53_{\pm0.01}$ | $0.78_{\pm0.00}$ | $0.29_{\pm0.01}$ | $0.71_{\pm0.01}$ | $0.39_{\pm0.03}$ |

| | | Antmaze umaze-diverse | Door cloned | Door expert | Door human | Hammer cloned |
|---|---|---|---|---|---|---|
| Abs. Error | IS | $\mathbf{0.14}_{\pm0.02}$ | $891_{\pm188}$ | $\mathbf{648}_{\pm122}$ | $870_{\pm173}$ | $7403_{\pm1126}$ |
| | VPM | $\mathbf{0.12}_{\pm0.03}$ | $1040_{\pm188}$ | $\mathbf{879}_{\pm182}$ | $862_{\pm163}$ | $7459_{\pm1114}$ |
| | Best DICE | $0.17_{\pm0.04}$ | $697_{\pm79}$ | $\mathbf{856}_{\pm134}$ | $1108_{\pm199}$ | $\mathbf{4169}_{\pm839}$ |
| | Doubly Robust | $\mathbf{0.11}_{\pm0.02}$ | $\mathbf{424}_{\pm73}$ | $1353_{\pm218}$ | $\mathbf{379}_{\pm65}$ | $6101_{\pm679}$ |
| | FQE (L2) | $\mathbf{0.11}_{\pm0.03}$ | $\mathbf{438}_{\pm81}$ | $1343_{\pm84}$ | $\mathbf{389}_{\pm60}$ | $\mathbf{5415}_{\pm558}$ |

| | | Hammer expert | Hammer human | Maze2d large | Maze2d medium | Maze2d umaze |
|---|---|---|---|---|---|---|
| Abs. Error | IS | $\mathbf{3052}_{\pm608}$ | $\mathbf{7352}_{\pm1118}$ | $45.61_{\pm10.43}$ | $61.29_{\pm7.78}$ | $50.20_{\pm9.16}$ |
| | VPM | $7312_{\pm1117}$ | $\mathbf{7105}_{\pm1107}$ | $44.10_{\pm10.69}$ | $60.30_{\pm8.37}$ | $62.81_{\pm8.40}$ |
| | Best DICE | $\mathbf{3963}_{\pm758}$ | $\mathbf{5677}_{\pm936}$ | $\mathbf{42.46}_{\pm9.66}$ | $58.97_{\pm9.57}$ | $\mathbf{21.95}_{\pm4.69}$ |
| | Doubly Robust | $\mathbf{3485}_{\pm590}$ | $\mathbf{5768}_{\pm751}$ | $\mathbf{22.94}_{\pm6.82}$ | $\mathbf{23.64}_{\pm4.96}$ | $76.93_{\pm4.42}$ |
| | FQE (L2) | $\mathbf{2950}_{\pm728}$ | $\mathbf{6000}_{\pm612}$ | $\mathbf{24.31}_{\pm6.56}$ | $\mathbf{35.11}_{\pm6.33}$ | $79.67_{\pm4.93}$ |

| | | Pen cloned | Pen expert | Pen human | Relocate cloned | Relocate expert |
|---|---|---|---|---|---|---|
| Abs. Error | IS | $1707_{\pm128}$ | $4547_{\pm222}$ | $3926_{\pm128}$ | $\mathbf{632}_{\pm215}$ | $2731_{\pm147}$ |
| | VPM | $2324_{\pm129}$ | $2325_{\pm136}$ | $\mathbf{1569}_{\pm215}$ | $\mathbf{586}_{\pm135}$ | $\mathbf{620}_{\pm214}$ |
| | Best DICE | $\mathbf{1454}_{\pm219}$ | $2963_{\pm279}$ | $4193_{\pm244}$ | $1347_{\pm485}$ | $\mathbf{1095}_{\pm221}$ |
| | Doubly Robust | $\mathbf{1323}_{\pm98}$ | $\mathbf{2013}_{\pm564}$ | $2846_{\pm200}$ | $\mathbf{412}_{\pm124}$ | $1193_{\pm350}$ |
| | FQE (L2) | $\mathbf{1232}_{\pm105}$ | $\mathbf{1057}_{\pm281}$ | $2872_{\pm170}$ | $\mathbf{439}_{\pm125}$ | $1351_{\pm393}$ |

| | | Relocate human | Ant expert | Ant medium | Ant medium-expert | Ant medium-replay |
|---|---|---|---|---|---|---|
| Abs. Error | IS | $638_{\pm217}$ | $\mathbf{605}_{\pm104}$ | $594_{\pm104}$ | $604_{\pm102}$ | $\mathbf{603}_{\pm101}$ |
| | VPM | $806_{\pm166}$ | $\mathbf{607}_{\pm108}$ | $570_{\pm109}$ | $604_{\pm106}$ | $\mathbf{612}_{\pm105}$ |
| | Best DICE | $4526_{\pm474}$ | $\mathbf{558}_{\pm108}$ | $495_{\pm90}$ | $\mathbf{471}_{\pm100}$ | $\mathbf{583}_{\pm110}$ |
| | Doubly Robust | $\mathbf{606}_{\pm116}$ | $\mathbf{584}_{\pm114}$ | $345_{\pm66}$ | $\mathbf{326}_{\pm66}$ | $\mathbf{421}_{\pm72}$ |
| | FQE (L2) | $\mathbf{593}_{\pm113}$ | $\mathbf{583}_{\pm122}$ | $345_{\pm64}$ | $\mathbf{319}_{\pm67}$ | $\mathbf{410}_{\pm79}$ |

| | | Ant random | Hopper expert | Hopper medium | Hopper random | Walker2d expert |
|---|---|---|---|---|---|---|
| Abs. Error | IS | $\mathbf{606}_{\pm103}$ | $\mathbf{106}_{\pm29}$ | $405_{\pm48}$ | $412_{\pm45}$ | $\mathbf{405}_{\pm62}$ |
| | VPM | $570_{\pm99}$ | $442_{\pm43}$ | $433_{\pm44}$ | $438_{\pm44}$ | $\mathbf{367}_{\pm68}$ |
| | Best DICE | $530_{\pm92}$ | $259_{\pm54}$ | $\mathbf{215}_{\pm41}$ | $\mathbf{122}_{\pm16}$ | $437_{\pm60}$ |
| | Doubly Robust | $404_{\pm106}$ | $426_{\pm99}$ | $\mathbf{307}_{\pm85}$ | $289_{\pm50}$ | $519_{\pm179}$ |
| | FQE (L2) | $398_{\pm111}$ | $282_{\pm76}$ | $\mathbf{283}_{\pm73}$ | $261_{\pm42}$ | $453_{\pm142}$ |

| | | Walker2d medium | Walker2d medium-expert | Walker2d medium-replay | Walker2d random | Median |
|---|---|---|---|---|---|---|
| Abs. Error | IS | $428_{\pm60}$ | $436_{\pm62}$ | $\mathbf{427}_{\pm60}$ | $\mathbf{430}_{\pm61}$ | $603.82$ |
| | VPM | $426_{\pm60}$ | $425_{\pm61}$ | $\mathbf{424}_{\pm64}$ | $\mathbf{440}_{\pm58}$ | $585.53$ |
| | Best DICE | $273_{\pm31}$ | $322_{\pm60}$ | $374_{\pm51}$ | $419_{\pm57}$ | $530.43$ |
| | Doubly Robust | $368_{\pm74}$ | $217_{\pm46}$ | $296_{\pm54}$ | $347_{\pm74}$ | $411.99$ |
| | FQE (L2) | $350_{\pm79}$ | $233_{\pm42}$ | $313_{\pm73}$ | $354_{\pm73}$ | $398.37$ |

| | | Halfcheetah expert | Halfcheetah medium-expert | Halfcheetah medium-replay | Halfcheetah random | Door cloned |
|---|---|---|---|---|---|---|
| Rank Corr. | Best DICE | $-0.44_{\pm 0.30}$ | $\mathbf{-0.08}_{\pm 0.35}$ | $\mathbf{-0.15}_{\pm 0.41}$ | $-0.70_{\pm 0.22}$ | $\mathbf{0.18}_{\pm 0.31}$ |
| | VPM | $\mathbf{0.18}_{\pm 0.35}$ | $-0.47_{\pm 0.29}$ | $\mathbf{-0.07}_{\pm 0.36}$ | $\mathbf{0.27}_{\pm 0.36}$ | $-0.29_{\pm 0.36}$ |
| | FQE (L2) | $\mathbf{0.78}_{\pm 0.15}$ | $\mathbf{0.62}_{\pm 0.27}$ | $\mathbf{0.26}_{\pm 0.37}$ | $-0.11_{\pm 0.41}$ | $\mathbf{0.55}_{\pm 0.27}$ |
| | IS | $0.01_{\pm 0.35}$ | $\mathbf{-0.06}_{\pm 0.37}$ | $\mathbf{0.59}_{\pm 0.26}$ | $\mathbf{-0.24}_{\pm 0.36}$ | $\mathbf{0.66}_{\pm 0.22}$ |
| | Doubly Robust | $\mathbf{0.77}_{\pm 0.17}$ | $\mathbf{0.62}_{\pm 0.27}$ | $\mathbf{0.32}_{\pm 0.37}$ | $\mathbf{-0.02}_{\pm 0.38}$ | $\mathbf{0.60}_{\pm 0.28}$ |

| | | Door expert | Hammer cloned | Hammer expert | Maze2d large | Maze2d medium |
|---|---|---|---|---|---|---|
| Rank Corr. | Best DICE | $-0.06_{\pm 0.32}$ | $\mathbf{0.35}_{\pm 0.38}$ | $-0.42_{\pm 0.31}$ | $\mathbf{0.56}_{\pm 0.21}$ | $-0.64_{\pm 0.23}$ |
| | VPM | $\mathbf{0.65}_{\pm 0.23}$ | $-0.77_{\pm 0.22}$ | $\mathbf{0.39}_{\pm 0.31}$ | $-0.26_{\pm 0.33}$ | $\mathbf{-0.05}_{\pm 0.39}$ |
| | FQE (L2) | $\mathbf{0.89}_{\pm 0.09}$ | $-0.15_{\pm 0.33}$ | $\mathbf{0.29}_{\pm 0.34}$ | $\mathbf{0.30}_{\pm 0.36}$ | $\mathbf{0.16}_{\pm 0.38}$ |
| | IS | $\mathbf{0.76}_{\pm 0.17}$ | $\mathbf{0.58}_{\pm 0.27}$ | $\mathbf{0.64}_{\pm 0.24}$ | $\mathbf{0.63}_{\pm 0.19}$ | $\mathbf{0.44}_{\pm 0.25}$ |
| | Doubly Robust | $\mathbf{0.76}_{\pm 0.13}$ | $-0.70_{\pm 0.20}$ | $\mathbf{0.49}_{\pm 0.31}$ | $\mathbf{0.31}_{\pm 0.36}$ | $\mathbf{0.41}_{\pm 0.35}$ |

| | | Pen expert | Relocate expert | Ant expert | Ant medium | Ant medium-expert |
|---|---|---|---|---|---|---|
| Rank Corr. | Best DICE | $-0.53_{\pm 0.30}$ | $-0.27_{\pm 0.34}$ | $\mathbf{-0.13}_{\pm 0.37}$ | $-0.36_{\pm 0.28}$ | $\mathbf{-0.33}_{\pm 0.40}$ |
| | VPM | $\mathbf{0.08}_{\pm 0.33}$ | $\mathbf{0.39}_{\pm 0.31}$ | $-0.42_{\pm 0.38}$ | $-0.20_{\pm 0.31}$ | $\mathbf{-0.28}_{\pm 0.28}$ |
| | FQE (L2) | $-0.01_{\pm 0.33}$ | $-0.57_{\pm 0.28}$ | $\mathbf{-0.13}_{\pm 0.32}$ | $\mathbf{0.65}_{\pm 0.25}$ | $\mathbf{0.37}_{\pm 0.35}$ |
| | IS | $-0.45_{\pm 0.31}$ | $\mathbf{0.52}_{\pm 0.23}$ | $\mathbf{0.14}_{\pm 0.41}$ | $-0.17_{\pm 0.32}$ | $\mathbf{-0.21}_{\pm 0.35}$ |
| | Doubly Robust | $\mathbf{0.52}_{\pm 0.28}$ | $-0.40_{\pm 0.24}$ | $\mathbf{-0.28}_{\pm 0.32}$ | $\mathbf{0.66}_{\pm 0.26}$ | $\mathbf{0.35}_{\pm 0.35}$ |

| | | Ant medium-replay | Ant random | Hopper expert | Hopper medium | Hopper random |
|---|---|---|---|---|---|---|
| Rank Corr. | Best DICE | $-0.24_{\pm 0.39}$ | $\mathbf{-0.21}_{\pm 0.35}$ | $\mathbf{-0.08}_{\pm 0.32}$ | $\mathbf{0.19}_{\pm 0.33}$ | $\mathbf{-0.13}_{\pm 0.39}$ |
| | VPM | $-0.26_{\pm 0.29}$ | $\mathbf{0.24}_{\pm 0.31}$ | $\mathbf{0.21}_{\pm 0.32}$ | $\mathbf{0.13}_{\pm 0.37}$ | $-0.46_{\pm 0.20}$ |
| | FQE (L2) | $\mathbf{0.57}_{\pm 0.28}$ | $\mathbf{0.04}_{\pm 0.33}$ | $\mathbf{-0.33}_{\pm 0.30}$ | $\mathbf{-0.29}_{\pm 0.33}$ | $\mathbf{-0.11}_{\pm 0.36}$ |
| | IS | $\mathbf{0.07}_{\pm 0.39}$ | $\mathbf{0.26}_{\pm 0.34}$ | $\mathbf{0.37}_{\pm 0.27}$ | $-0.55_{\pm 0.26}$ | $\mathbf{0.23}_{\pm 0.34}$ |
| | Doubly Robust | $\mathbf{0.45}_{\pm 0.32}$ | $\mathbf{0.01}_{\pm 0.33}$ | $-0.41_{\pm 0.27}$ | $\mathbf{-0.31}_{\pm 0.34}$ | $\mathbf{-0.19}_{\pm 0.36}$ |

| | | Walker2d expert | Walker2d medium | Walker2d medium-expert | Walker2d medium-replay | Walker2d random |
|---|---|---|---|---|---|---|
| Rank Corr. | Best DICE | $\mathbf{-0.37}_{\pm 0.27}$ | $\mathbf{0.12}_{\pm 0.38}$ | $\mathbf{-0.34}_{\pm 0.34}$ | $\mathbf{0.55}_{\pm 0.23}$ | $\mathbf{-0.19}_{\pm 0.36}$ |
| | VPM | $\mathbf{0.17}_{\pm 0.32}$ | $\mathbf{0.44}_{\pm 0.21}$ | $\mathbf{0.49}_{\pm 0.37}$ | $-0.52_{\pm 0.25}$ | $\mathbf{-0.42}_{\pm 0.34}$ |
| | FQE (L2) | $\mathbf{0.35}_{\pm 0.33}$ | $\mathbf{-0.09}_{\pm 0.36}$ | $\mathbf{0.25}_{\pm 0.32}$ | $-0.19_{\pm 0.36}$ | $\mathbf{0.21}_{\pm 0.31}$ |
| | IS | $\mathbf{0.22}_{\pm 0.37}$ | $\mathbf{-0.25}_{\pm 0.35}$ | $\mathbf{0.24}_{\pm 0.33}$ | $\mathbf{0.65}_{\pm 0.24}$ | $\mathbf{-0.05}_{\pm 0.38}$ |
| | Doubly Robust | $\mathbf{0.26}_{\pm 0.34}$ | $\mathbf{0.02}_{\pm 0.37}$ | $\mathbf{0.19}_{\pm 0.33}$ | $-0.37_{\pm 0.39}$ | $\mathbf{0.16}_{\pm 0.29}$ |

| | | Median |
|---|---|---|
| Rank Corr. | Best DICE | $-0.19$ |
| | VPM | $-0.05$ |
| | FQE (L2) | $0.21$ |
| | IS | $0.23$ |
| | Doubly Robust | $0.26$ |

| Regret@1 | | Halfcheetah expert | Halfcheetah medium | Halfcheetah medium-expert | Halfcheetah medium-replay | Halfcheetah random |
|---|---|---|---|---|---|---|
| | Best DICE | $0.32_{\pm0.40}$ | $0.82_{\pm0.29}$ | $0.38_{\pm0.37}$ | $0.30_{\pm0.07}$ | $0.81_{\pm0.30}$ |
| | VPM | $0.14_{\pm0.09}$ | $0.33_{\pm0.19}$ | $0.80_{\pm0.34}$ | $0.25_{\pm0.09}$ | $0.12_{\pm0.07}$ |
| | Doubly Robust | $0.11_{\pm0.08}$ | $0.37_{\pm0.15}$ | $0.14_{\pm0.07}$ | $0.33_{\pm0.18}$ | $0.31_{\pm0.10}$ |
| | FQE (L2) | $0.12_{\pm0.07}$ | $0.38_{\pm0.13}$ | $0.14_{\pm0.07}$ | $0.36_{\pm0.16}$ | $0.37_{\pm0.08}$ |
| | IS | $0.15_{\pm0.08}$ | $0.05_{\pm0.05}$ | $0.73_{\pm0.42}$ | $0.13_{\pm0.10}$ | $0.31_{\pm0.11}$ |

| Regret@1 | | Antmaze large-diverse | Antmaze large-play | Antmaze medium-diverse | Antmaze medium-play | Antmaze umaze |
|---|---|---|---|---|---|---|
| | Best DICE | $0.54_{\pm0.34}$ | $0.96_{\pm0.13}$ | $0.04_{\pm0.11}$ | $0.09_{\pm0.10}$ | $0.69_{\pm0.39}$ |
| | VPM | $0.88_{\pm0.27}$ | $0.45_{\pm0.30}$ | $0.14_{\pm0.10}$ | $0.03_{\pm0.08}$ | $0.62_{\pm0.32}$ |
| | Doubly Robust | $0.83_{\pm0.30}$ | $0.93_{\pm0.21}$ | $0.05_{\pm0.07}$ | $0.17_{\pm0.31}$ | $0.42_{\pm0.36}$ |
| | FQE (L2) | $0.93_{\pm0.25}$ | $1.00_{\pm0.03}$ | $0.16_{\pm0.10}$ | $0.05_{\pm0.19}$ | $0.41_{\pm0.35}$ |
| | IS | $0.39_{\pm0.26}$ | $0.71_{\pm0.20}$ | $0.14_{\pm0.09}$ | $0.18_{\pm0.06}$ | $0.86_{\pm0.06}$ |

| Regret@1 | | Antmaze umaze-diverse | Door cloned | Door expert | Door human | Hammer cloned |
|---|---|---|---|---|---|---|
| | Best DICE | $0.42_{\pm0.28}$ | $0.65_{\pm0.45}$ | $0.37_{\pm0.27}$ | $0.10_{\pm0.27}$ | $0.67_{\pm0.48}$ |
| | VPM | $0.63_{\pm0.32}$ | $0.81_{\pm0.33}$ | $0.03_{\pm0.03}$ | $0.69_{\pm0.24}$ | $0.72_{\pm0.39}$ |
| | Doubly Robust | $0.79_{\pm0.14}$ | $0.11_{\pm0.08}$ | $0.05_{\pm0.07}$ | $0.05_{\pm0.09}$ | $0.78_{\pm0.38}$ |
| | FQE (L2) | $0.64_{\pm0.37}$ | $0.11_{\pm0.06}$ | $0.03_{\pm0.03}$ | $0.05_{\pm0.08}$ | $0.36_{\pm0.39}$ |
| | IS | $0.22_{\pm0.36}$ | $0.02_{\pm0.07}$ | $0.01_{\pm0.04}$ | $0.45_{\pm0.40}$ | $0.03_{\pm0.15}$ |

| Regret@1 | | Hammer expert | Hammer human | Maze2d large | Maze2d medium | Maze2d umaze |
|---|---|---|---|---|---|---|
| | Best DICE | $0.24_{\pm0.34}$ | $0.04_{\pm0.08}$ | $0.15_{\pm0.08}$ | $0.44_{\pm0.05}$ | $0.03_{\pm0.07}$ |
| | VPM | $0.04_{\pm0.07}$ | $0.18_{\pm0.29}$ | $0.66_{\pm0.10}$ | $0.24_{\pm0.24}$ | $0.06_{\pm0.12}$ |
| | Doubly Robust | $0.09_{\pm0.09}$ | $0.46_{\pm0.23}$ | $0.21_{\pm0.16}$ | $0.27_{\pm0.14}$ | $0.03_{\pm0.07}$ |
| | FQE (L2) | $0.05_{\pm0.04}$ | $0.46_{\pm0.23}$ | $0.20_{\pm0.14}$ | $0.31_{\pm0.14}$ | $0.03_{\pm0.07}$ |
| | IS | $0.01_{\pm0.04}$ | $0.19_{\pm0.30}$ | $0.16_{\pm0.23}$ | $0.15_{\pm0.15}$ | $0.02_{\pm0.12}$ |

| Regret@1 | | Pen cloned | Pen expert | Pen human | Relocate cloned | Relocate expert |
|---|---|---|---|---|---|---|
| | Best DICE | $0.12_{\pm0.08}$ | $0.33_{\pm0.20}$ | $0.04_{\pm0.09}$ | $0.96_{\pm0.18}$ | $0.97_{\pm0.07}$ |
| | VPM | $0.36_{\pm0.18}$ | $0.25_{\pm0.13}$ | $0.28_{\pm0.12}$ | $0.11_{\pm0.29}$ | $0.76_{\pm0.23}$ |
| | Doubly Robust | $0.13_{\pm0.06}$ | $0.05_{\pm0.07}$ | $0.09_{\pm0.08}$ | $0.18_{\pm0.27}$ | $0.98_{\pm0.08}$ |
| | FQE (L2) | $0.12_{\pm0.07}$ | $0.11_{\pm0.14}$ | $0.07_{\pm0.05}$ | $0.29_{\pm0.42}$ | $1.00_{\pm0.06}$ |
| | IS | $0.14_{\pm0.09}$ | $0.31_{\pm0.10}$ | $0.17_{\pm0.15}$ | $0.63_{\pm0.41}$ | $0.18_{\pm0.14}$ |

| Regret@1 | | Relocate human | Ant expert | Ant medium | Ant medium-expert | Ant medium-replay |
|---|---|---|---|---|---|---|
| | Best DICE | $0.97_{\pm0.11}$ | $0.62_{\pm0.15}$ | $0.43_{\pm0.10}$ | $0.60_{\pm0.16}$ | $0.64_{\pm0.13}$ |
| | VPM | $0.77_{\pm0.18}$ | $0.88_{\pm0.22}$ | $0.40_{\pm0.21}$ | $0.32_{\pm0.24}$ | $0.72_{\pm0.43}$ |
| | Doubly Robust | $0.17_{\pm0.15}$ | $0.43_{\pm0.22}$ | $0.12_{\pm0.18}$ | $0.37_{\pm0.13}$ | $0.05_{\pm0.09}$ |
| | FQE (L2) | $0.17_{\pm0.14}$ | $0.43_{\pm0.22}$ | $0.12_{\pm0.18}$ | $0.36_{\pm0.14}$ | $0.05_{\pm0.09}$ |
| | IS | $0.63_{\pm0.41}$ | $0.47_{\pm0.32}$ | $0.61_{\pm0.18}$ | $0.46_{\pm0.18}$ | $0.16_{\pm0.23}$ |

| Regret@1 | | Ant random | Hopper expert | Hopper medium | Hopper random | Walker2d expert |
|---|---|---|---|---|---|---|
| | Best DICE | $0.50_{\pm0.29}$ | $0.20_{\pm0.08}$ | $0.18_{\pm0.19}$ | $0.30_{\pm0.15}$ | $0.35_{\pm0.36}$ |
| | VPM | $0.15_{\pm0.24}$ | $0.13_{\pm0.10}$ | $0.10_{\pm0.14}$ | $0.26_{\pm0.10}$ | $0.09_{\pm0.19}$ |
| | Doubly Robust | $0.28_{\pm0.15}$ | $0.34_{\pm0.35}$ | $0.32_{\pm0.32}$ | $0.41_{\pm0.17}$ | $0.06_{\pm0.07}$ |
| | FQE (L2) | $0.28_{\pm0.15}$ | $0.41_{\pm0.20}$ | $0.32_{\pm0.32}$ | $0.36_{\pm0.22}$ | $0.06_{\pm0.07}$ |
| | IS | $0.56_{\pm0.22}$ | $0.06_{\pm0.03}$ | $0.38_{\pm0.28}$ | $0.05_{\pm0.05}$ | $0.43_{\pm0.26}$ |

| Regret@1 | | Walker2d medium | Walker2d medium-expert | Walker2d medium-replay | Walker2d random | Median |
|---|---|---|---|---|---|---|
| | Best DICE | $0.27_{\pm0.43}$ | $0.78_{\pm0.27}$ | $0.18_{\pm0.12}$ | $0.39_{\pm0.33}$ | 0.38 |
| | VPM | $0.08_{\pm0.06}$ | $0.24_{\pm0.42}$ | $0.46_{\pm0.31}$ | $0.88_{\pm0.20}$ | 0.28 |
| | Doubly Robust | $0.25_{\pm0.09}$ | $0.30_{\pm0.12}$ | $0.68_{\pm0.23}$ | $0.15_{\pm0.20}$ | 0.25 |
| | FQE (L2) | $0.31_{\pm0.10}$ | $0.22_{\pm0.14}$ | $0.24_{\pm0.20}$ | $0.15_{\pm0.21}$ | 0.24 |
| | IS | $0.70_{\pm0.39}$ | $0.13_{\pm0.07}$ | $0.02_{\pm0.05}$ | $0.74_{\pm0.33}$ | 0.18 |

### A.2.3 SCATTER PLOTS

Finally, we present scatter plots plotting the true returns of each policy against the estimated returns. Each point on the plot represents one evaluated policy.

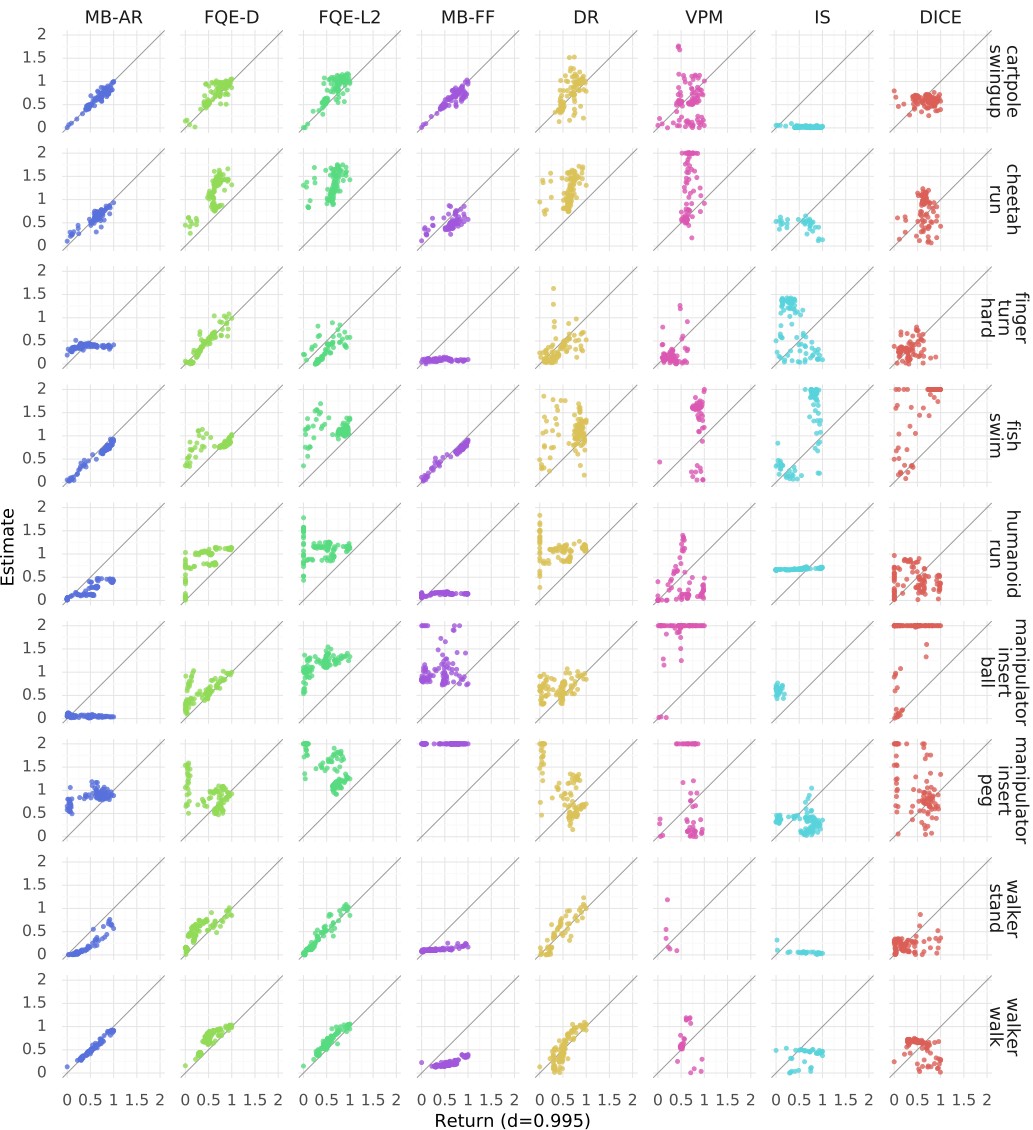

Figure A.8: Scatter plots of estimate vs ground truth return for each baseline on each task in DOPE RL Unplugged.

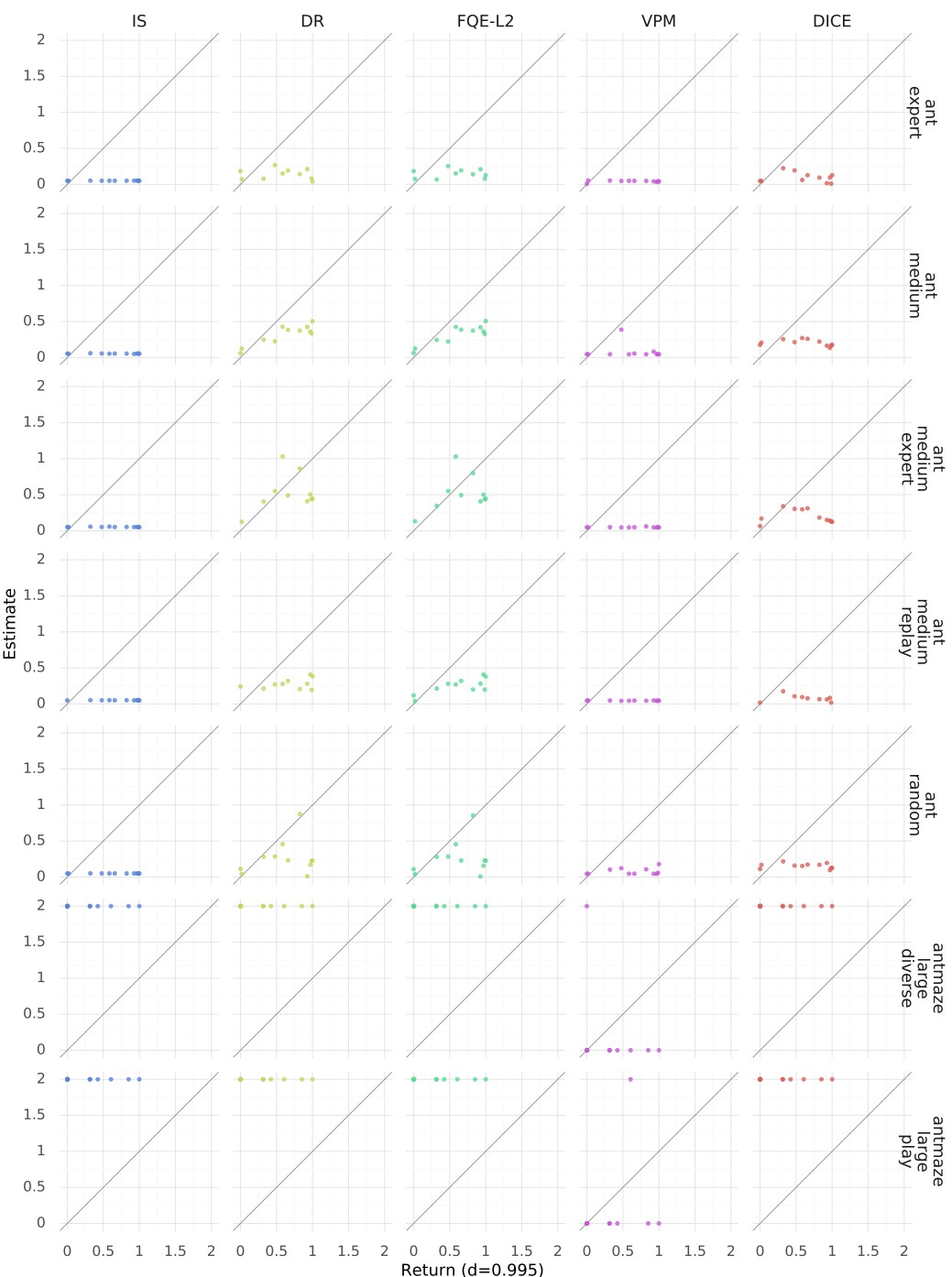

Figure A.9: Scatter plots of estimate vs ground truth return for each baseline on each task in DOPE D4RL (part 1).

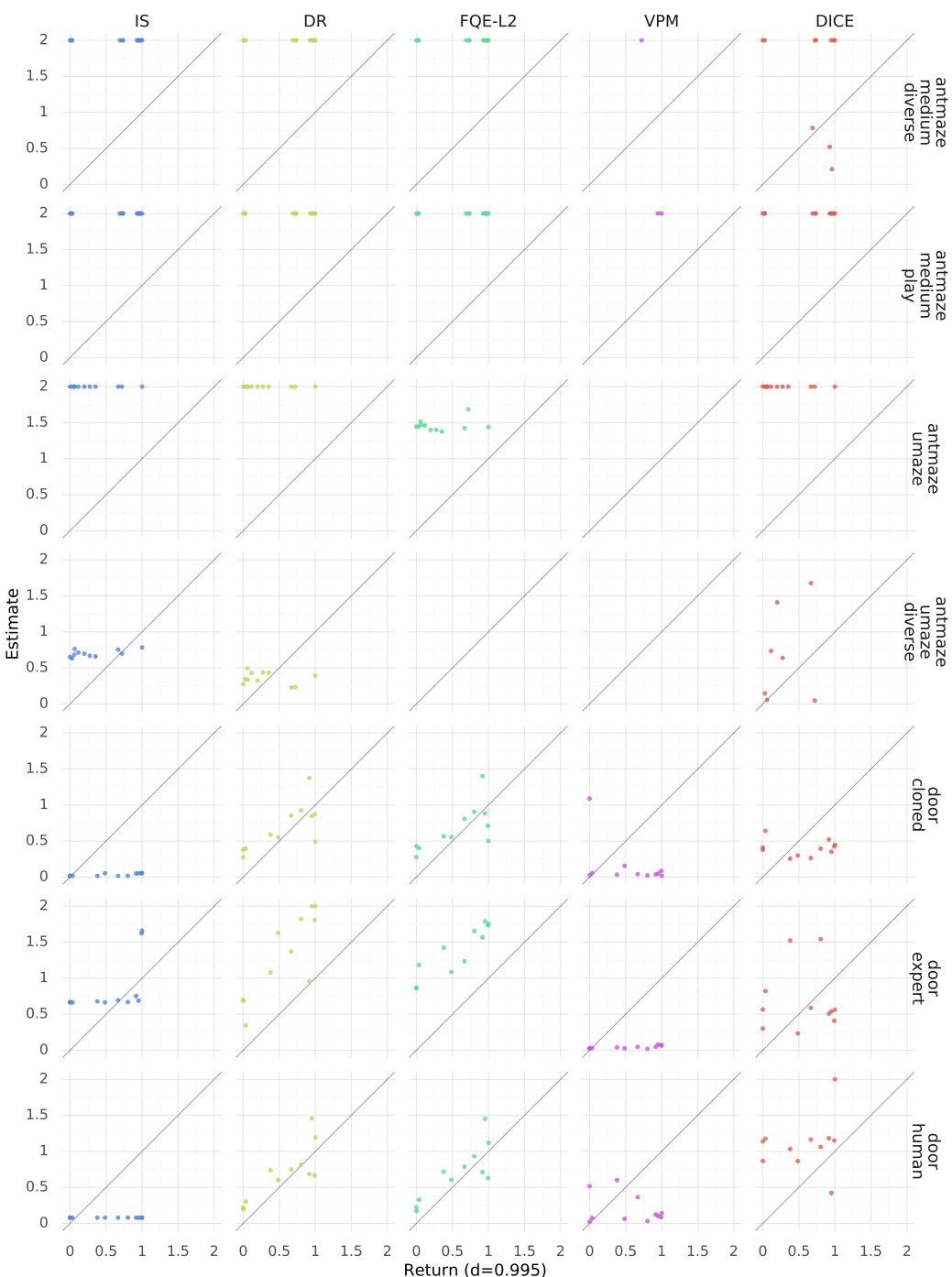

Figure A.10: Scatter plots of estimate vs ground truth return for each baseline on each task in DOPE D4RL (part 2).

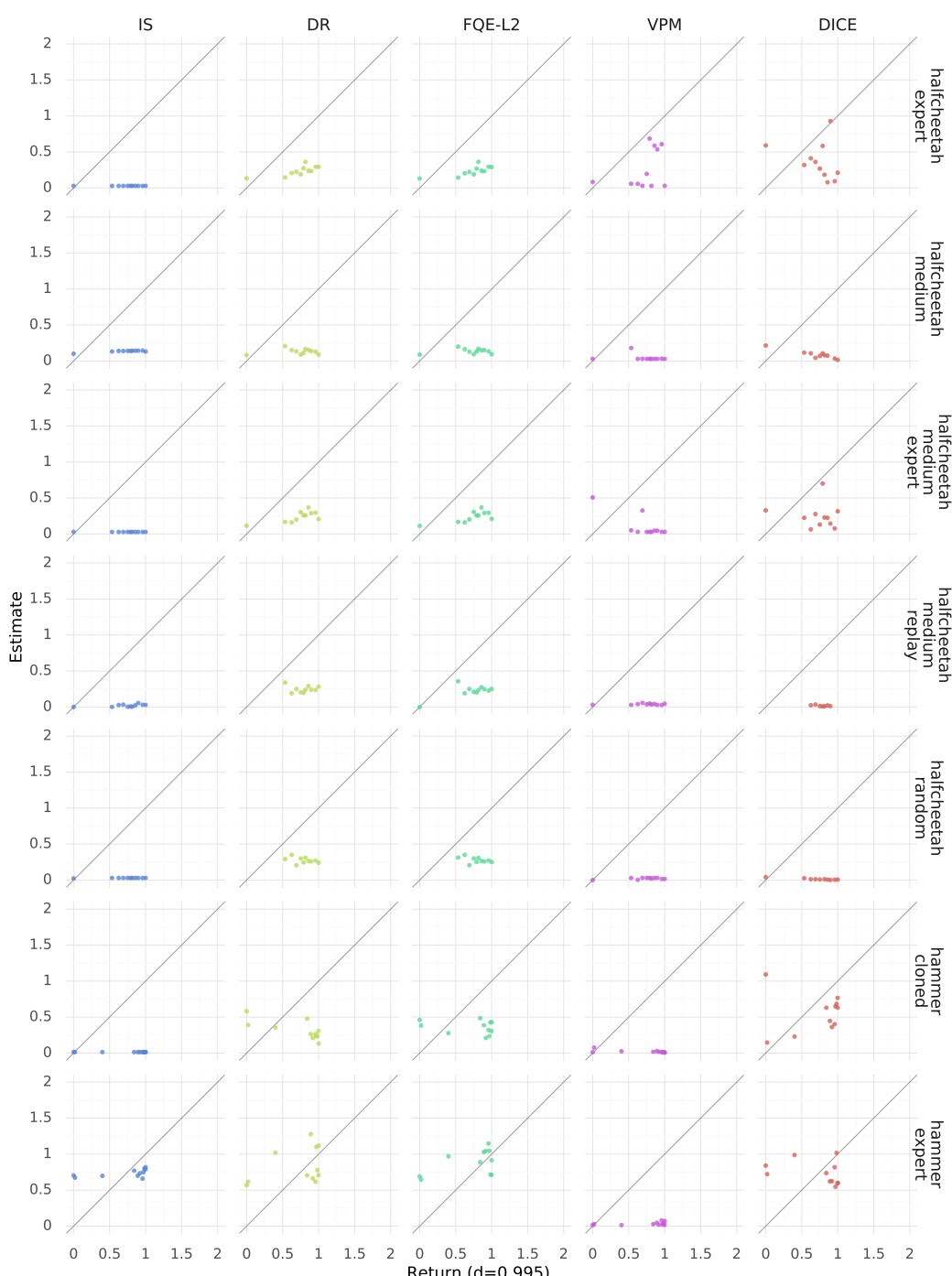

Figure A.11: Scatter plots of estimate vs ground truth return for each baseline on each task in DOPE D4RL (part 3).

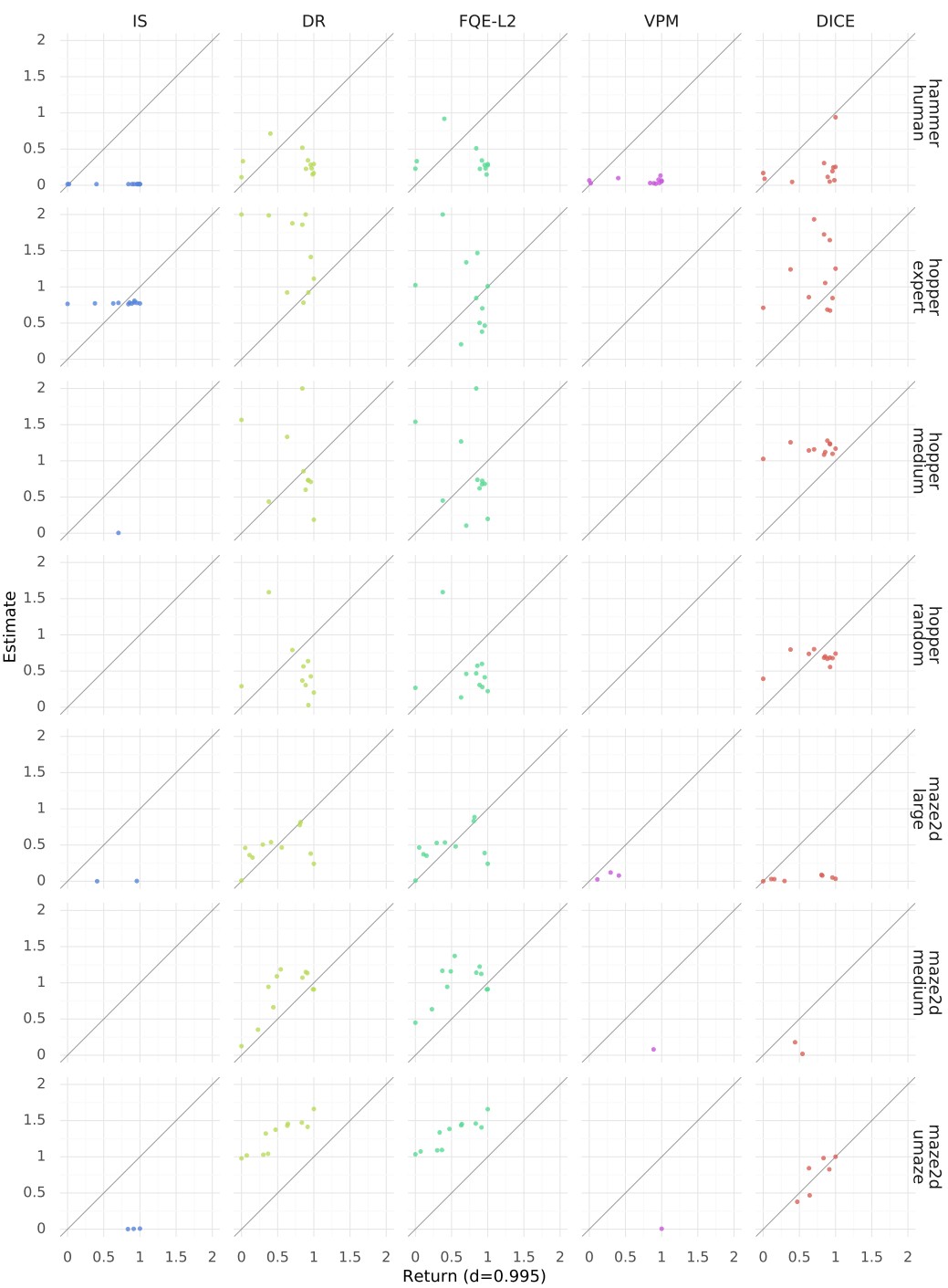

Figure A.12: Scatter plots of estimate vs ground truth return for each baseline on each task in DOPE D4RL (part 4).

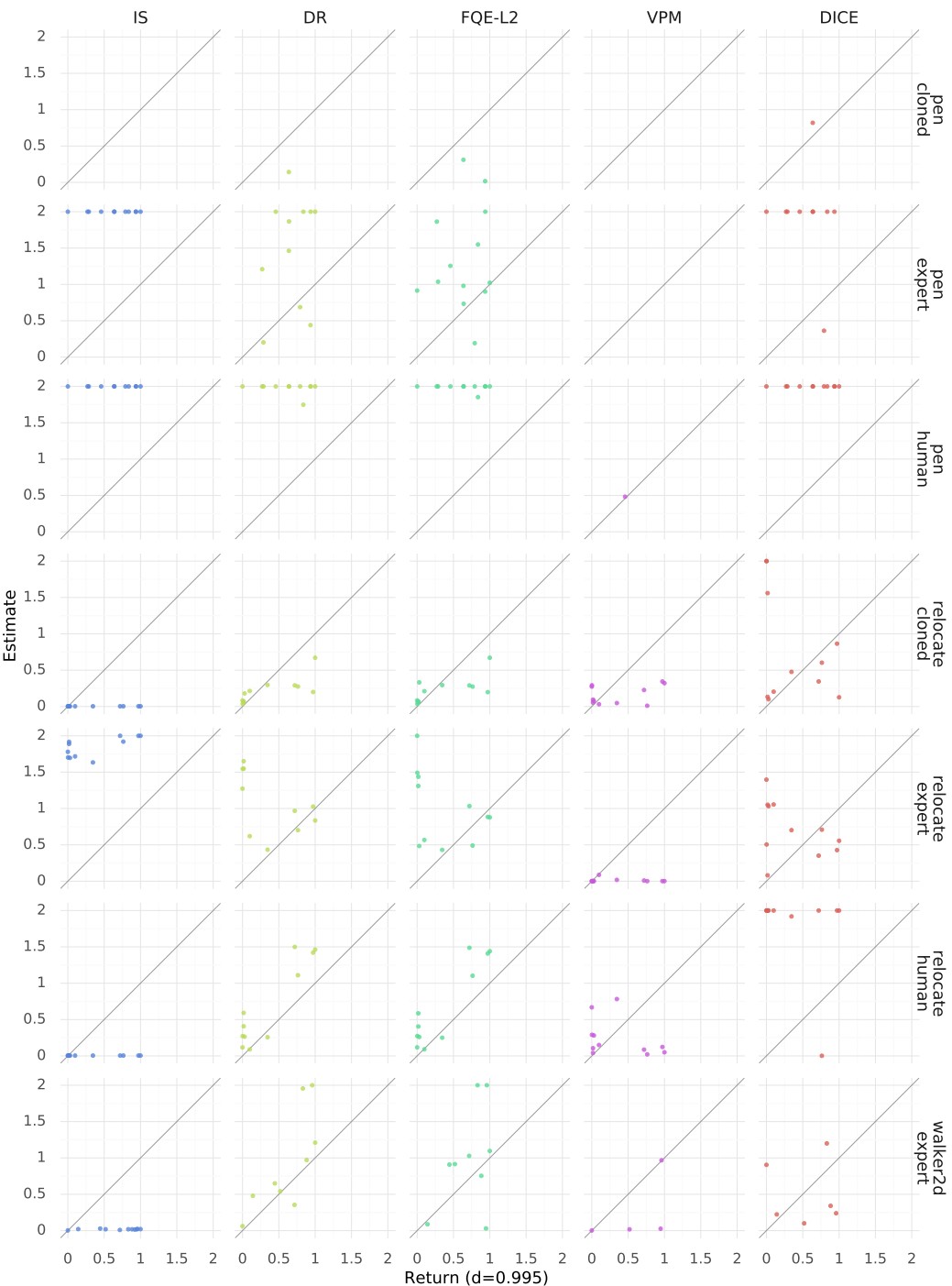

Figure A.13: Scatter plots of estimate vs ground truth return for each baseline on each task in DOPE D4RL (part 5).

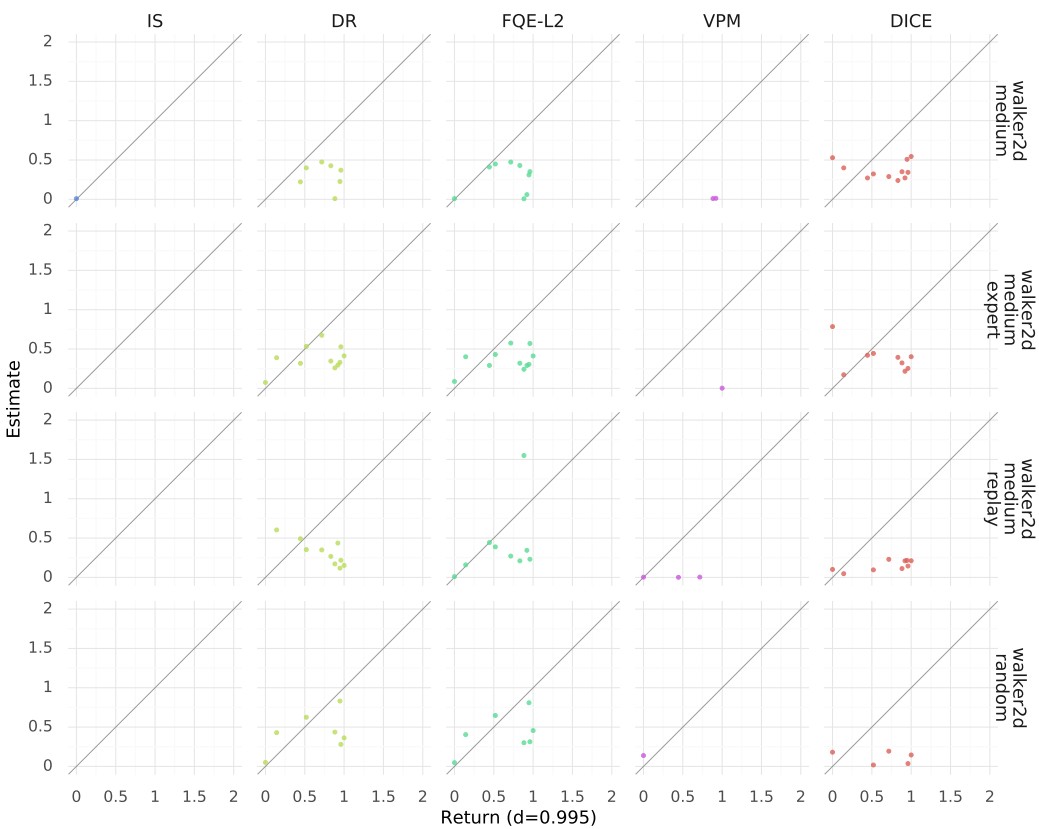

Figure A.14: Scatter plots of estimate vs ground truth return for each baseline on each task in DOPE D4RL (part 6).

