# OpenReview forum: "Benchmarks for Deep Off-Policy Evaluation"
_ICLR.cc/2021/Conference — ICLR 2021 Poster_

### Official Review · AnonReviewer1 · 2020-10-22
**Great stuff, I plan on using this myself, but no connection to representation learning as presented**

**Rating:** 7
**Confidence:** 4

**Review:**

This work describes a set of benchmarks for offline policy evaluation and selection.  The area is important and well-motivated in the introduction.

The work builds upon existing benchmark tasks for offline RL.  However, the increment is meaningful.  By providing the triad of data, trained policies, and online evaluation results, authors have isolated the offline evaluation component, controlled for ancillary aspects, and facilitated experimentation.  In other words, experiments are both easier and more useful.

The work as presented makes little attempt to connect with representation learning which is the focus of the ICLR conference.  It could be improved in this regard, if authors make connections between estimation and representation (e.g., how are model-based estimators related to or employing representation learning?   what about model-free estimators that have to fit a propensity model to the behaviour policy?  how, if at all, do the baselines described in section 5.1 utilize representation learning, or how could they be improved via representation learning techniques?)

The name of the work is highly unfortunate.  There is nothing deep about the benchmark in the representation learning sense, and the use of the term deep contributes to the general degradation of the term into a meaningless yet popular adjective.  Unfortunately, les jeux sons faits ... do better next time!

---

> ### Author Response · Authors · 2020-11-21
> **Author response to reviewer 1**
>
> Thank you for your valuable feedback.
>
> > "connect with representation learning which is the focus of the ICLR conference [...] name of the work is highly unfortunate"
>
> Thank you. We should make this more clear. Many of the OPE baselines make use of improvements in neural function approximation (e.g., nonlinearities, good initialization, activation normalization, etc) and improvements focused on function approximation for reinforcement learning (e.g., target networks, double DQN, etc). The non-trivial dimensionality of the continuous state and action spaces requires generalization to evaluate novel policies from a fixed offline dataset. We expect other ideas from the deep learning and RL communities to further improve the performance on these benchmarks. Accordingly, we believe the ICLR conference is well suited for this paper and including Deep in the name is appropriate. We will update the paper to reflect this more clearly.

---

### Official Review · AnonReviewer2 · 2020-10-25
**This paper gives a standardized benchmark for off-policy evaluation, which contains fixed logged datasets and various target policies to cover wide range of challenges.**

**Rating:** 7
**Confidence:** 4

**Review:**

1. Significance
- I view this paper has high significance level. As more and more OPE methods arise, papers rely on different datasets and metrics, which makes them hard to compare, and especially see the "success" and "failure" mode of this method. We need a benchmark dataset on this domain and this paper moves an important step towards this by providing standardized logged dataset, and various policies in different challenging, complex domains.

2. Novelty
- There is a very similar work by Voloshin et.al (2019), which gives the empirical evaluation of popular OPE methods across various domains. Though not providing the logged data, they give the logging policy to logged the data, I am concerned about the novelty of this work compared with Voloshin et.al (2019). It would be great if the author could discuss the novelty/improvement over that work.

3. Quality & Clarity
- This paper has a great structure, metrics/choice of datasets/design of policies are well motivated and clearly stated, it is a very easy to read paper.

- This work gives a clear description of the logged datasets/policies/and the performance of various off-policy method. However, I feel it would be great if the authors could also think, and if possible make the following discussions along with this paper:

(1). it would be great if the authors could give some statistics to measure the property of the datasets, such as a measure of the missing support of logging policy, how difference are the logging policy and target policy?

(2). I know the authors discuss some properties of the environment in words, such as sparse/dense rewards, stochasticity of the environment, etc. It would be great to summarize these environment info into a single table.

(3). A very important class  of estimators in OPE is these kind of interpolated estimators between value-based and weighting-based, which contains a trade-off of bias-variance, such as the work by Phillip and Emma (2016). It also shows in Voloshin et.al (2019)'s empirical evaluations and shows very promising performance. However, it is missing in this paper and it would be excited to see how it works in the D4RL tasks.

(4). A further extension from (1) is that the authors did a great job in doing lots of empirical evaluations in various domains. I saw aggregated performance in main paper and individual task performance is listed in Appendix. It would be great to classify them into groups which depends on  the MDP property, the metric measuring difference of logging/target policy, and etc... This will definitely give some further directions like which method perform well in which setting. This is also shows in Voloshin et.al (2019)'s paper but kind of missing in this work.

(5). One difference compared with Voloshin et.al (2019) is that it seems in the logged dataset, they know the propensity. However, I feel the dataset released along the paper does not have propensity. Will is possible to also add some datasets with propensity included?

Overall, I feel this paper addressing an important problem, however, I feel there is still room for improvement to make them have more impact. I am willing to adjust my score if the authors address some of these issues.

Ref:
1.  Philip S. Thomas, Emma Brunskill  Data-Efficient Off-Policy Policy Evaluation for Reinforcement Learning.
2. Cameron Voloshin, Hoang M. Le, Nan Jiang, Yisong Yue Empirical Study of Off-Policy Policy Evaluation for Reinforcement Learning.

---

> ### Author Response · Authors · 2020-11-21
> **Author response to reviewer 2**
>
> Thank you for your comments. Please let us know if your concerns are addressed.
>
> > “I am concerned about the novelty of this work compared with Voloshin et.al (2019).”
>
> We updated our paper to make the differences between the proposed benchmark and Voloshin et al. (2019) clearer.
>
> Voloshin et al. (2019) present their primary contribution as an empirical comparison of different OPE methods across a range of situations (e.g., representation mismatch, horizon length, policy mismatch, bad behavior policy estimates), and provide guidance for each. They perform this evaluation on environments with discrete actions including: tabular grid world and graph environments, mountain car, and an Atari environment, namely the Enduro environment. They conclude with important future work problems, and our work directly addresses two of them:
> - OPE for continuous actions, which pose significant fundamental challenges for many OPE methods: All of our tasks are continuous control.
> - Missing data coverage: We don't assume full support of the evaluation policy by the behavior policy data. In fact we design our datasets and policies to ensure a range of missing data coverage (See the description of DOPE D4RL in the Section 4). Crucially, this makes good representation learning a key component for any successful OPE method. For instance, the importance sampling baseline is not competitive on these tasks.
>
> Beyond estimating value, our work provides a large set of evaluation policies trained by offline RL and online RL, making it possible to measure how OPE algorithms perform at ranking these policies. We are not aware of prior standardized benchmarks that enable this. Offline policy selection is a particularly appealing use case for applications.
>
> Finally, we note that because we use existing open-source offline RL datasets as our behavior data, using our benchmarks is more straightforward, reproducible, and standardized. That said,  Voloshin et al and our benchmark are complementary, and we invite future research on OPE to consider both benchmarks for evaluation.
>
> > “I know the authors discuss some properties of the environment in words, such as sparse/dense rewards, stochasticity of the environment, etc. It would be great to summarize these environment info into a single table.”
>
> We have updated Table 1 with an additional row that details which properties correspond to which tasks.
>
> > “ A very important class of estimators in OPE is these kind of interpolated estimators between value-based and weighting-based, which contains a trade-off of bias-variance, such as the work by Phillip and Emma (2016). It also shows in Voloshin et.al (2019)'s empirical evaluations and shows very promising performance. However, it is missing in this paper and it would be excited to see how it works in the D4RL tasks.”
>
> We agree. The weighted doubly-robust policy evaluation method (labeled DR in our evaluations) is based on Thomas and Brunskill (2016). But it would be good to include MAGIC as well. Time permitting, we will add that baseline to the final version of the paper.
>
> > “ One difference compared with Voloshin et.al (2019) is that it seems in the logged dataset, they know the propensity. However, I feel the dataset released along the paper does not have propensity. Will is possible to also add some datasets with propensity included?”
>
> Adding propensity is difficult because the behavior policies used to generate some of the D4RL/RLUnplugged datasets are a mixture of many policies followed by a filtering step to keep the datasets small and challenging. We believe not including the propensity can help encourage future work to focus on *behavior-agnostic* OPE methods, which is typical in many practical applications, where offline datasets of interactions are not amenable to logging propensities (e.g., when the behavior policy is a mix of ML and hard-coded business logic). For the methods that require propensities, we estimate the propensities with behavior cloning. We are happy to share the estimated policy propensities learned by behavioral cloning on our offline datasets if you think that is helpful.

---

> > ### Comment · AnonReviewer2 · 2020-11-25
> > **Response to authors**
> >
> > Thanks authors for the response! It clears most of my concerns, though some are left as "to do task" in the final version of the paper. I am willing to increase my score and hopefully these changes will be seen in the final version.

---

### Official Review · AnonReviewer4 · 2020-11-05
**The review of benchmarks for deep off-policy evaluation**

**Rating:** 6
**Confidence:** 5

**Review:**

This article proposes a benchmark of off-policy evaluation, which provides different metrics for policy ranking, evaluation and selection. Offline metrics are provided by evaluating the value function of logged data, and then evaluating absolute error, rank correlation and regret. Verify the effectiveness of different offline evaluation methods. This article provides two evaluation scenarios, one is DOPE RL unplugged, and the other is D4RL. In the experiment, the author verified the benchmark proposed in this article in the MuJoCo environment to evaluate the effectiveness of different offline evaluation methods.

The paper’s key strengths:

1.	I think the research direction of this article is very meaningful, because many scenarios in the real world only have offline data, which requires a simulated environment or other offline evaluation methods for strategy evaluation and model selection. This is very crucial for model improvement and evaluation. And the author proposed absolute error and rank correlation in the benchmark to evaluate the strategy. I think this is very innovative.
2.	The author has applied a large number of offline evaluation methods in the experiment to verify in different metrics, and in the appendix has conducted experiments and verifications for many MuJoCo scenarios. I think the experimental results are very rich and very confirmatory.

The paper’s key weaknesses:
1.	The biggest problem with this article is that the contribution of the article is insufficient and lacks originality. This article proposes a benchmark for off-policy evaluation and verifies different OPE methods, but this article does not compare with other similar benchmarks to verify whether the benchmark proposed in this article is effective.
2.	In the experimental part, this paper verifies different metrics for different OPE methods. However, in Figure 4 and Figure 5, the different methods in the two sets of benchmarks proposed in this article are quite different in different OPE methods. I hope the author can give some comments on the differences between the two sets of evaluation methods.
3.	The author uses the value function in formula 1 to estimate the effect of the strategy. I doubt this method. Because of the attenuation factor here, I think it has an impact on the final value calculated by different methods. Because bellman equation is only an estimate of the value of a certain state, not an absolute strategy benefit, I hope the author will give some explanations here.

---

> ### Author Response · Authors · 2020-11-13
> **Author response**
>
> Thank you for the comments, and we address the issues raised in your review below. Please let us know if our responses address your concerns, or if there are any additional clarifications needed.
>
> > "... this article does not compare with other similar benchmarks"
>
> We updated our paper to make the differences between our benchmark and other recent OPE benchmarks clearer in the paper. Overall, to the best of our knowledge, there are no widely used OPE benchmarks with tasks comparable to the high-dimensional, continuous action problems as presented in this paper, though we would be happy to add any other references that the reviewer might suggest. The most similar work that we are familiar with is Voloshin et al. (2019). Voloshin et al. (2019) present their primary contribution as an empirical comparison of different OPE methods across a range of situations (e.g., representation mismatch, horizon length, policy mismatch, bad behavior policy estimates), and provide guidance for each. They perform this evaluation on environments with discrete actions including: tabular grid world and graph environments, mountain car, and an Atari environment, namely the Enduro environment. They conclude with important future work problems, and our work directly addresses two of them:
> - OPE for continuous actions, which pose significant fundamental challenges for many OPE methods: All of our tasks are continuous control.
> - Missing data coverage: We don't assume full support of the evaluation policy by the behavior policy data. In fact we design our datasets and policies to ensure a range of missing data coverage (See the description of DOPE D4RL in the Section 4). Crucially, this makes good representation learning a key component for any successful OPE method.
>
> Beyond estimating value, our work provides a large set of evaluation policies trained by offline RL and online RL, making it possible to measure how OPE algorithms perform at ranking these policies. We are not aware of prior standardized benchmarks that enable this. Offline policy selection is a particularly appealing use case for applications.
>
> Finally, we note that because we use existing open-source offline RL datasets as our behavior data, using our benchmarks is more straightforward, reproducible, and standardized. That said, these benchmarks are complementary, and we invite future research on OPE to consider both benchmarks for evaluation.
>
> > “ In the experimental part, this paper verifies different metrics for different OPE methods. However, ... different methods in the two sets of benchmarks proposed in this article are quite different in different OPE methods.”
>
> We assume you are referring to the difference in the relative performance of different OPE methods with respect to the different metrics we propose. This is a good question, and please clarify if this is not the case.
>
> We find that the rank order of OPE methods in terms of Regret@1 and Rank correlation is very similar in Figures 4 and 5, but the order of OPE methods according to Absolute error is sometimes different. This is not too surprising, since in theory an OPE technique that ranks policies correctly does not have to estimate the precise value of each policy correctly. In practice, we find that the importance sampling baseline suffers from this issue, and even though it ranks policies reasonably in D4RL, the value estimates are generally poor. See magenta scatter plots on pages 27-32, where the r-values are acceptable, but the value estimates are poor. We note that for certain applications one may care about the correctness of absolute value estimates of policies while in others one cares about the relative value and ranking of policies, hence both metrics are valuable. We will include this discussion in the paper.
>
> > “The author uses the value function in formula 1 to estimate the effect of the strategy. [...]”
>
> Thank you for pointing out this confusion. We measure expected discounted return in the MDP, which is the expected value under the initial state distribution (i.e., $\mathbb{E}_{s_0,\pi}[r_0+\gamma r_1 + \gamma^2 r_2 + \dots ]$ -- note this does not use any learned functions). We updated Eq. 1 to reflect this.
>
> We would like to emphasize that this ground truth value is independent of the method used to estimate the expected discounted return, and therefore it does not have an “impact on the final value calculated by different methods”. Please let us know if you have additional concerns.

---

### Official Review · AnonReviewer5 · 2020-11-06
**It's a Necessary Benchmark Paper**

**Rating:** 6
**Confidence:** 4

**Review:**

## Summary

The paper proposes an off-policy evaluation dataset (DOPE) for various control tasks.
The authors include various baselines for off-policy evaluation and evaluate them with additional metrics to MSE, namely regret@k and rank correlation.

## Writeup

The writeup is good. It is clear what the "features" of the proposed benchmark are, and what problems the benchmark addresses that the other benchmarks didn't address before. I don't really like that in different images form  dm-control and gym environments pop up in the paper. Since these environments as self-standing were not the contribution of the paper I would remove these images, unless they are used to show a specific point. The space should be used rather to illustrate the point of the paper, that additional metrics are needed for evaluating algorithms / better benchmark is needed.

## Pros

I think that using additional evaluation metrics for off-policy are indeed necessary. The authors show that not one single metric summarizes performance over all of the metrics. This is going to allow researchers to do a more thorough analysis of offline reinforcement learning algorithms.

## Cons

Instead of using the space for "nice pictures", use it to actually support the claims of the paper, maybe shifting some figures from the appendix to the main text would be more informative.
The paper is not making any improvement over the methods or suggesting any reason why there is such a disparity in performance between different metrics, as such, there are no deeper insights.
A large chunk of the paper is previous work.

## Comments

In general, the experimental evaluation is thorough (there are a lot of experiments in the appending with a lot of tables and figures).

To me it is a bit questionable to which extent this data might count as high-dimensional and complicated, in comparison to real-world of robotics, this data is still quite low dimensional. It would be nice to see this dataset contain image data for learning from image input.

I am highly suspicious about the way the data is generated for the ant-maze environment. Specifically, you use expert data for ant-maze-large because DAPG didn't work. Please specify where does the expert data come from.

I didn't try to run the code, but I looked at it. I didn't find the code for the algorithms themselves. I would expect to have the training code to reproduce the experiments. With the current code I can only evaluate saved policies?

Can you comment on what kind of normalization was done on the reward function? Ideally it should be normalized across all environments in the benchmark, such that performance is comparable.

Maybe also writing down the equations for regret@k and rank correlation would be helpful. It is sometimes simpler to understand the metric once one sees the actual equation.

---

> ### Author Response · Authors · 2020-11-21
> **Author response to reviewer 5**
>
> Thank you for your valuable feedback. We will address the issues you raised below.
>
> > "Instead of using the space for 'nice pictures', use it to actually support the claims of the paper, maybe shifting some figures from the appendix to the main text would be more informative."
>
> This is a good idea. Originally, we were concerned readers might not be familiar with some of the environments and tasks. and hoped images may highlight the nature of the tasks. But we agree it is better to move these images to the appendix, and move some results back into the main body. We will incorporate this comment into the final version.
>
> > "It would be nice to see this dataset contain image data for learning from image input."
>
> We agree that benchmarking OPE methods on datasets with image inputs is valuable. We plan to add this to the benchmark in a future version, but cannot guarantee to have it in time for ICLR
>
> > "I am highly suspicious about the way the data is generated for the ant-maze environment. Please specify where does the expert data come from."
>
> The expert data for antmaze-large is generated by collecting additional trajectories that reach the goal using a high-level waypoint planner in conjunction with a low-level goal-conditioned policy. This is the same method used to generate the dataset (See Sec. 5 in the D4RL paper), except we collected additional trajectories which reach the goal rather than navigation to random goals. We have updated the paper to clarify this.
>
> > "I didn't find the code for the algorithms themselves."
>
> We agree that an implementation of OPE algorithms should be open sourced. We obtained the code for the algorithms from the cited papers via personal communication. We are currently actively working with the authors of those cited papers to open source the algorithm code in the coming months.
>
> > "Can you comment on what kind of normalization was done on the reward function? Ideally it should be normalized across all environments in the benchmark, such that performance is comparable."
>
> Good point. For DOPE RL Unplugged, the environments already have normalized rewards. For DOPE D4RL, we normalized by the max discounted return achieved by any policy for that task.
> We mention this briefly in the caption for Figure 5, but will expand upon this in the appendix.
>
> > "Maybe also writing down the equations for regret@k and rank correlation would be helpful. It is sometimes simpler to understand the metric once one sees the actual equation."
>
> Good suggestion. We will add this in the final version.

---

### Decision · Program_Chairs · 2021-01-07
**Final Decision**

**Decision:**

Accept (Poster)

**Comment:**

This paper proposes benchmark tasks for offline policy evaluation.  The proposed benchmark tasks evaluate the policy with batch data with respect to three metrics, including the standard mean squared error. The paper also evaluate several baseline offline reinforcement learning methods with the benchmark tasks, which will serve as standard baselines.  All of the reviewers are in favor of the paper.